🔓 | Antimicrobial Chemotherapy | Research Article

# *In vitro* and *ex vivo* proteomics of *Mycobacterium marinum* biofilms and the development of biofilm-binding synthetic nanobodies

Milka Marjut Hammarén,[1,2] Hanna Luukinen,[1] Alina Sillanpää,[1] Kim Remans,[2] Karine Lapouge,[2] Tânia Custódio,[3,4,5] Christian Löw,[3,4,5] Henna Myllymäki,[1] Toni Montonen,[1] Markus Seeger,[6] Joseph Robertson,[7,8] Tuula A. Nyman,[7,8] Kirsi Savijoki,[1,9] Mataleena Parikka[1]

**ABSTRACT** The antibiotic-tolerant biofilms present in tuberculous granulomas add an additional layer of complexity when treating mycobacterial infections, including tuberculosis (TB). For a more efficient treatment of TB, the biofilm forms of mycobacteria warrant specific attention. Here, we used *Mycobacterium marinum* (Mmr) as a biofilm-forming model to identify the abundant proteins covering the biofilm surface. We used biotinylation/streptavidin-based proteomics on the proteins exposed at the Mmr biofilm matrices *in vitro* to identify 448 proteins and *ex vivo* proteomics to detect 91 Mmr proteins from the mycobacterial granulomas isolated from adult zebrafish. *In vitro* and *ex vivo* proteomics data are available via ProteomeXchange with identifiers PXD033425 and PXD039416, respectively. Data comparisons pinpointed the molecular chaperone GroEL2 as the most abundant Mmr protein within the *in vitro* and *ex vivo* proteomes, while its paralog, GroEL1, with a known role in biofilm formation, was detected with slightly lower intensity values. To validate the surface exposure of these targets, we created in-house synthetic nanobodies (sybodies) against the two chaperones and identified sybodies that bind the mycobacterial biofilms *in vitro* and those present in *ex vivo* granulomas. Taken together, the present study reports a proof-of-concept showing that surface proteomics *in vitro* and *ex vivo* proteomics combined is a valuable strategy to identify surface-exposed proteins on the mycobacterial biofilm. Biofilm surface–binding nanobodies could be eventually used as homing agents to deliver biofilm-targeting treatments to the sites of persistent biofilm infection.

**IMPORTANCE** With the currently available antibiotics, the treatment of TB takes months. The slow response to treatment is caused by antibiotic tolerance, which is especially common among bacteria that form biofilms. Such biofilms are composed of bacterial cells surrounded by the extracellular matrix. Both the matrix and the dormant lifestyle of the bacterial cells are thought to hinder the efficacy of antibiotics. To be able to develop faster-acting treatments against TB, the biofilm forms of mycobacteria deserve specific attention. In this work, we characterize the protein composition of Mmr biofilms in bacterial cultures and in mycobacteria extracted from infected adult zebrafish. We identify abundant surface-exposed targets and develop the first sybodies that bind to mycobacterial biofilms. As nanobodies can be linked to other therapeutic compounds, in the future, they can provide means to target therapies to biofilms.

**KEYWORDS** *Mycobacterium*, biofilm, surface proteome, synthetic nanobody libraries, nanobody, biofilm-targeted therapy

Address correspondence to Milka Marjut Hammarén, milka.hammaren@tuni.fi.

Milka Marjut Hammarén and Hanna Luukinen contributed equally to this article. The more senior author is listed first.

Kirsi Savijoki and Mataleena Parikka contributed equally to this article.

The authors declare no conflict of interest.

See the funding table on p. 22.

*M*ycobacterium tuberculosis (Mtb) is currently the deadliest bacterial pathogen in the world (1) and has long been known to form antibiotic-tolerant biofilms *in vitro* (2, 3). Biofilms are difficult and slow to fully eradicate with antibiotics (4). Tolerance arises from both the physical protection provided by an extracellular matrix surrounding the bacteria as well as the altered phenotypes of the bacterial cells embedded within the matrix (4). Antibiotic tolerance has also been shown to support the development of genetic antibiotic resistance, thereby exacerbating treatment challenges (5). Therefore, treatments that can target antibiotic-tolerant biofilms could also hinder the development of genetic antibiotic resistance—an enormous global health problem. According to the World Health Organization, 48% of previously treated infections are multidrug-resistant tuberculosis (TB). The treatment success rate for resistant strains is only 59% (1).

Recent studies have taken a closer look into TB granulomas harboring antibiotic-tolerant populations. The caseous center of granulomas in the rabbit model of TB was shown to contain bacteria that are extremely antibiotic tolerant (6, 7). Recently, in a landmark paper by Chakraborty and colleagues, the presence of antibiotic-tolerant mycobacteria was shown to coincide with the presence of cellulose in the biofilm matrix inside of granulomas isolated from the lungs of mice, monkeys, and humans (8). The disassembly of the matrix with an enzymatic cellulase treatment was shown to reduce the antibiotic tolerance of mycobacteria *in vivo* (8). Thus, alternative treatment modalities beyond antibiotics specifically targeting antibiotic-tolerant biofilms can open new horizons in the fight against TB and other biofilm infections as well as provide better strategies to reduce antibiotic use, thereby hindering the further development and spread of genetic antibiotic resistance.

Nanobodies are small, single-domain antibodies originally identified in camelids. Compared to traditional immunoglobulin antibodies, nanobodies are small (~15 kDa) and have higher stability, low immunogenicity, and better tissue penetration. Nanobodies can also easily be linked to other functional therapeutic molecules, making them promising tools for specialized treatment delivery in the battle against hard-to-reach/hard-to-treat diseases (9, 10). Similar to existing cancer-targeting antibody-linked therapeutics, we envision treatment-delivering nanobodies could be developed against the extracellular proteins of bacterial biofilms for the purpose of concentrating alternative treatments to bacterial biofilm lesions. As nanobodies cannot penetrate cells, the ideal nanobody targets would be the abundant surface-exposed proteins on the biofilms.

Recently, we started to explore the extracellular composition of *Mycobacterium marinum* (Mmr) biofilms (11). We chose Mmr as our biofilm model as it forms mature biofilms *in vitro* (12, 13) that are antibiotic-tolerant (11) and is a well-established model pathogen for TB (14–17). Mmr causes a TB-like disease in zebrafish, which share central aspects with human TB (18, 19). Typical is the formation of caseous necrotic granulomas and the replication of the spectrum of disease states ranging from active progressive to chronic infection with dormant bacteria or reactivated disease (14, 18). Our previous work on cell surface–shaving proteomics provided a systems-level view of the biofilm matrix proteome and how the protein profile changes over time (11). However, to reliably elucidate which of the biofilm matrix proteins are the most promising binding targets for nanobodies, in this study, we created a parallel *in vitro* data set using cell surface protein biotinylation combined with liquid chromatography-mass spectrometry/mass spectrometry identification. In addition, we provide the first insights into the mycobacterial *in vivo* proteome of Mmr granulomas extracted from adult zebrafish. The extensive comparative analysis of these three overlapping data sets allowed us to reliably identify surface-exposed biofilm proteins in mycobacterial infection.

To validate the surface exposure of selected hits and to take the first steps toward therapeutic delivery strategies, we also developed nanobodies against two mycobacterial chaperones, GroEL1 and GroEL2. Instead of using camelid immunizations for the generation of binders against mycobacterial biofilms, we used the synthetic nanobody (sybody) libraries developed by Zimmermann and colleagues (20, 21). We generated

sybodies against two proteins identified from our proteomic analyses and showed their binding to Mmr biofilms. When chemically linked to fluorescent cargo, these nanobodies could be used to label biofilm *in vitro* and *ex vivo* in granulomas. Our results demonstrate the potential of surface proteomic strategies to identify nanobody-binding targets on biofilms and the capacity of using biofilm-targeting nanobodies to bind cargo onto biofilms in granulomas.

## MATERIALS AND METHODS

### Bacteria and culture conditions

*M. marinum* (ATCC927) and the avirulent *M. tuberculosis* strain H37Ra (ATCC25177) were used in the present study. Mmr cells were cultured at 29°C and Mtb at 37°C. Mycobacterial biofilms were first cultured on a 7H10 agar plate supplemented with 10% (vol/vol) of OADC (oleic acid, albumin, dextrose, catalase) enrichment (Thermo Fisher Scientific, New Hampshire, USA) and 0.5% (vol/vol) glycerol (Sigma-Aldrich, Missouri, USA) for 1 week (Mmr) or 3 weeks (Mtb). After the pre-culturing, the bacterial mass was transferred to a 7H9 medium supplemented with 10% of ADC (albumin, dextrose, catalase) enrichment (Fisher Scientific) to obtain an optical density (OD600) of 0.1. The culturing was continued for a further 2–5 weeks in containers sealed with laboratory film.

### Biotinylation proteomics of *in vitro* biofilms

#### *Protein biotinylation*

To biotinylate the proteins on intact and lyzed Mmr biofilms, 5-week-old biofilms containing both the pellicle and submerged biofilm cells were pooled together, centrifuged, and resuspended in BupH-PBS (Thermo Fisher Scientific, New Hampshire, USA) and stored on ice until biotinylation. To produce the total lysate protein samples, the biofilm cells were lysed by bead beating (100-µm glass beads at 6.5 m/s twice for 40 s with dry ice) and sonicated for 10 min in a water bath in the presence of 20 mg/mL lysozyme (Sigma-Aldrich) for 2 hours at 37°C. The residual cell debris was centrifuged to obtain cell-free extract containing both the cytoplasmic and the biofilm matrix/cell wall–associated proteins. Then, both the intact and lyzed Mmr biofilms were biotinylated with sulfo-NHS-LC-biotin (Pierce, Illinois, USA) using 1 mg of biotin per 150 mg of the sample in BupH-PBS at room temperature (RT) for 30 min with gentle agitation. After the incubation, 10 mg/mL of glycine was added to terminate the reaction. The labeled, intact biofilm cells were resuspended in a 600-µL urea lysis buffer composed of 140 mM NaCl, 20 mM $Na_2HPO_4$, 7 M urea, 0.05% (vol/vol) Tween 20, and 0.1% (wt/vol) deoxycholic acid (pH 7.2) for lyzing the cells by bead beating, as described above. The disrupted samples were centrifuged (12,000$g$ for 10 min) to remove insoluble substances, and excess biotin was removed from the lysates by dialysis using 3-kDa Slide-A-Lyzer cassettes in BupH-PBS.

The affinity capture of the biotinylated bacterial proteins was performed in Safe-Seal low-binding tubes (BioScience, Utah, USA) using magnetic streptavidin-coated C1 dynabeads (Invitrogen, California, USA) in 150 mM NaCl, 20 mM $Na_2HPO_4$, 1.75 M urea, 0.05% (vol/vol) Tween 20 (Sigma-Aldrich), and 0.05% (wt/vol) CHAPS at pH 7.3. The samples were washed three times with the same buffer; three times with 150 mM NaCl, 20 mM $Na_2HPO_4$, and 0.05% (vol/vol) Tween 20 at pH 7.2; and once with 50 mM $NH_4HCO_3$ at pH 7.8 and flash-frozen in 50 mM $NH_4HCO_3$.

#### *On-bead Tryptic digestion and liquid chromatography tandem mass spectrometry (LC-MS/MS) identification of biotinylated proteins*

The streptavidin beads with captured biotin-labeled proteins were resuspended in fresh 50 mM $NH_4HCO_3$, then reduced using 10 mM dithiotreitol (DTT), and alkylated using 15 mM iodoacetamide. Protein samples were digested using 1 µg trypsin (Promega,

Wisconsin, USA) and incubated overnight at 37°C. Following digestion, the samples were acidified and desalted using homemade C18 stage tips. Peptides were eluted from the stage tips using 50% acetonitrile (ACN)/0.1% formic acid (FA), dried to completion by speed vacuum, and resuspended in 0.1% FA. LC-MS/MS analysis was performed using a nanoElute nanoflow ultrahigh pressure LC system (Bruker Daltonics, Bremen, Germany) coupled to a timsTOF fleX mass spectrometer (Bruker Daltonics) with CaptiveSpray nanoelectrospray ion source (Bruker Daltonics). The peptides were separated using a 60-min gradient at a flow rate of 300 nL/min. The timsTOF fleX was operated in PASEF mode, and the data-dependent acquisition was performed using 10 PASEF MS/MS scans per cycle with a near 100% duty cycle.

## *Ex vivo* proteomics of the mycobacterial granulomas

### *Zebrafish housing, Mmr infections, and granuloma extraction*

Adult 5- to 10-month-old female AB wild-type zebrafish (*Danio rerio*) were used in the experiments. The fish were housed in flow-through water-circulation systems with a 14-hour/10-hour light/dark cycle. For zebrafish infections, *M. marinum* ATCC927 carrying the pTEC27 plasmid that expresses the tdTomato fluorescent protein, Addgene plasmid no. 30182 (Addgene, Massachusetts, USA) was cultured in the Middlebrook 7H9 medium with ADC enrichment (Fisher Scientific) with 0.2% Tween 80 (Sigma-Aldrich) for 4 days, diluted to an OD600 of 0.07, and cultured for a further 2 days until an OD600 of ~0.4 was reached. The bacteria were harvested by centrifuging for 3 min at 10,000$g$ and then resuspended and diluted in sterile 1× PBS (phosphate-buffered saline) with 0.3 mg/mL of Phenol Red (Sigma-Aldrich). The fish were anesthetized with 0.02% 3-aminobenzoic acid ethyl ester (pH 7.0) (Sigma-Aldrich), and a total of 5 µL of bacterial suspension (63±9 cfu/fish) was injected intraperitoneally with an Omnican 100 30 G insulin needle (Braun, Melsungen, Germany). At 8 wpi, 10 zebrafish were euthanized with 0.04% 3-aminobenzoic acid ethyl ester (pH 7.0) (Sigma-Aldrich). Red fluorescent mycobacterial granulomas were carefully dissected from the zebrafish ovaries under a stereomicroscope and the NightSea light with a 600-nm filter (Electron Microscopy Science, Massachusetts, USA) using sharp forceps. Ten granulomas were collected per tube, frozen on dry ice, and stored at −80°C until preparation for proteomics.

### *Extraction of mycobacterial proteins and LC-MS/MS identification*

Extraction of proteins from mycobacterial granulomas for on-bead aggregation/digestion was conducted as follows. The granulomas in 10 replicates (each with 10 individual granulomas) were mixed with 150 µL of 0.1% RapiGest in 50 mM (wt/vol) in Tris-HCl (pH 8.0) and transferred into FastPrep-24 (MP Biomedicals, California, USA) tubes with six ceramic beads. Mycobacterial granulomas with beads were subjected to mechanistic beating with the speed set at level 6 for three cycles (30 s each) in a FastPrep-24 (MP Biomedicals) with cooling on ice between the pulses to soften/disrupt the granulomas without homogenizing the mycobacterial biofilms. Then, 150 µL of 0.4% RapiGest (wt/vol) was added onto the softened granulomas, and the suspensions were incubated at RT for 2 hours with frequent mixing. Proteins solubilized into RapiGest were separated from the beads by centrifugation (16,000$g$, 5 min at 20°C), and the protein concentration was measured with NanoDrop 2000/2000c spectrophotometers (Thermo Fisher Scientific, Vantaa, Finland). Protein samples were sent to the microparticle-assisted sample preparation prior to label-free quantification (LFQ) identifications using the recently reported method (22) with the following modifications. Briefly, acetonitrile at 70% (vol/vol) was used to aggregate proteins in RapiGest and 10 µL of MAgReSyn Amine beads (20 mg/mL; ReSyn Biosciences, Gauteng, South Africa) to bind the protein aggregates. For on-bead digestion, 50 mM of Tris-HCL (pH 8.0) was added to the beads, and the proteins were reduced with 10 mM DTT for 30 min at 37°C, alkylated with 20 mM iodoacetamide for 30 min at RT in the dark and quenched with 20 mM DTT. Trypsin/Lys-C Mix (mass-spec grade; Promega) was added to each sample at a 25:1 protein:protease

ratio (wt/wt). The samples were gently mixed and incubated overnight at 37°C. The beads were separated by magnet; the supernatants with the released peptides were transferred into new tubes and acidified with 0.6% trifluoroacetic acid (TFA) (vol/vol); and the peptides were desalted using C18 StageTips. Peptides were analyzed using the nanoElute nanoflow ultrahigh pressure LC system combined with timsTOF fleX mass spectrometer using CaptiveSpray nanoelectrospray ionization with analysis parameters as described above.

## Mass spectrometry data analysis

Raw MS files generated from both the *in vitro* and *ex vivo* proteomic approaches were searched with the MaxQuant software (version 2.0.1.0 for *in vitro* data, v.1.6.1.0 for *ex vivo* data) (23, 24) using the UniProt Mmr database (*in vitro* data) or a database composed of both zebrafish (*D. rerio*, Proteome ID: UP000000437, 25,707 proteins, https://www.uniprot.org/proteomes/UP000000437) and Mmr (Proteome ID: UP000001190, 5,418 proteins, https://www.uniprot.org/proteomes/UP000001190) protein sequences (*ex vivo* data). Carbamidomethyl (C) was set as a fixed modification, while methionine oxidation and protein *N*-terminal acetylation were set as a variable modification. The first search peptide tolerance of 20 ppm and main search peptide tolerance of 10 ppm were used. Trypsin without the proline restriction enzyme option was used, with two allowed miscleavages. The minimal unique + razor peptide number was set to 1, and the false discovery rate (FDR) was set to 0.01 (1%) for peptide and protein identification. Generation of reversed sequences was selected to assign the FDR. The mass spectrometry proteomics data have been deposited into the ProteomeXchange Consortium via the PRIDE partner repository (25, 26). The mass spectrometry proteomics data have been deposited to the ProteomeXchange Consortium via the PRIDE partner repository with the data set identifier PXD033425 for the *in vitro* data and PXD039416 for the *ex vivo* data.

### Proteome bioinformatics and statistics

Protein sequences for all identified proteins were retrieved at the UniProt Knowledgebase (UniProtKB) composed of two sections, UniProtKB/Swiss-Prot and UniProtKB/TrEMBL. To determine the pIs and molecular weights (MWs) of the identified proteins, the protein sequences were submitted to EMBOSS Pepstats (27) analyses at https://www.ebi.ac.uk/Tools/seqstats/emboss_pepstats/. The presence of possible protein secretion motifs (SPI, SPII, TATP, non-classical) for all identified proteins was obtained with SignalP 6.0 (https://services.healthtech.dtu.dk/service.php?SignalP) (28). Helices/TMDs were determined with the TMHMM Server v. 2.0 at https://services.healthtech.dtu.dk/service.php?TMHMM-2.0 (29, 30) for the identified proteins. Perseus v. 2.0.3.1 (24) was used to compare the biotinylated and non-biotinylated LFQ data sets. For pairwise comparisons, a paired t-test and $P < 0.05$ with a minimum of two valid identifications in at least one of the groups were used to indicate statistically significant LFQ value changes. For indicating proteins with significantly higher abundances on the Mmr biofilm matrices compared to the planktonic cell surfaces *in vitro*, we used the recently published LFQ data for statistical comparisons. For multivariate analyses (hierarchical clustering and PCA), the missing values within the LFQ data were replaced by imputed values from the normal distribution (width = 0.3, downshift = 1.8) and then normalized (z-score).

## Expression and purification of GroEL chaperones

The full-length sequences of Mmr GroEL1 and GroEL2 were ordered as synthetic genes with an *N*-terminal His6-3C-AVI-TEV fusion tag and codon optimized for recombinant expression in *E. coli*. The construct design allowed the simultaneous production of biotinylated and tag-free forms of the target proteins. The constructs containing the synthetic genes subcloned into pET24a expression vectors were ordered from GeneArt (Regensburg, Germany). Chemically competent *E. coli* BL21(DE3) cells were transformed

with the pET24a plasmids together with a BirA-encoding pACYC84 plasmid to allow the *in vivo* biotinylation of the AVI-tagged GroEL1 and GroEL2 proteins. The transformed bacteria were grown in LB on a shaker at 37°C overnight under selective pressure with chloramphenicol (34 µg/mL) and kanamycin (30 µg/mL). The following day, cultures were diluted 1:100 in antibiotic-containing LB with 0.5% glucose and cultured at 37°C with shaking at 200 rpm to an OD600 of 0.6. Subsequently, the cultures were induced with 0.5 mM Isopropyl β-D-1-thiogalactopyranoside (IPTG), and 50 µM of biotin was added. The bacteria were then cultured overnight at 18°C at 200 rpm and collected by 30 min centrifugation at 4°C and 4,000$g$. The cell pellets were resuspended in buffer A (50 mM Tris-HCl pH 8, 500 mM NaCl, 20 mM imidazole, 10% glycerol) with the following additives: 1 mg/mL lysozyme, 0.01 mg/mL DNAse I, 2 mM $MgCl_2$, and protease inhibitors cOmplete EDTA-free protease inhibitor cocktail (Roche, Basel, Switzerland). The cells were lyzed using a microfluidizer device. The cleared supernatants were incubated with QIAGEN Ni-NTA agarose beads on rotation for 2 hours at 4°C to bind the His-tagged proteins. The beads were washed with a lysis buffer, and the His-tagged GroEL1 and GroEL2 proteins were eluted in buffer A supplemented with 500 mM imidazole. The eluted proteins were divided into two aliquots and dialyzed overnight at 4°C against 50 mM Tris-HCl pH 8.0, 300 mM NaCl, 20 mM imidazole, and 10% glycerol with simultaneous protease treatment with either His6-3C protease or His6-TEV protease (ratio 1:40) to generate AVI-tagged biotinylated proteins or tag-free proteins, respectively. The dialyzed proteins were subjected to a reverse NiNTA to remove the proteases and uncleaved products using the same buffer conditions as described in the affinity capture step of the target proteins. The proteins were further purified and analyzed by size exclusion chromatography (SEC) columns Superdex 200 10/300 in 50 mM Tris-HCl pH 8, 150 mM NaCl, 10% glycerol, and 1 mM tris(2-carboxyethyl)phosphine (TCEP). The elution fractions with the monodisperse product were pooled, concentrated to 2 mg/mL, aliquoted, flash-frozen in liquid $N_2$, and stored at −80°C.

## Sybody selections

Sybody selections were carried out with the fully synthetic screening platform, as described previously (20, 21). The synthetic library was produced by the Seeger Laboratory in Zurich University and delivered in the form of mRNA. Three different libraries (concave, loop, and convex) differing in the length of the CDR3 loop of the nanobody were used. The biotinylated target protein (GroEL1 or GroEL2) was immobilized, and the selections were carried out in three phases, starting with ribosome display and followed by two rounds of phage display. In the second round of phage display, the low-affinity binders were washed off with a competition buffer containing the target protein at 5 µM. The progress of the selections was followed by quantitative PCR. A biotinylated maltose-binding protein was used as the negative control to assess the enrichment of binders after each phage display round.

## ELISA

After the three selection rounds, the sybody sequences were FX-cloned into expression vectors to produce His-myc-tagged sybodies. Per library, 94 clones were selected and produced in the periplasm of *E. coli* MC1061 F-. For the enzyme-linked immunosorbent assay (ELISA), the sybodies were first immobilized via Protein A + anti-Myc antibody, after which 50 nM of biotinylated GroEL1 or GroEL2 protein was added. Unspecific mannose-binding protein (MBP) was used as a negative control. The level of binding was quantified with streptavidin-horse radish peroxidase (HRP) and 3,3',5, 5' - tetramethylbenzidine (TMB) in 50 mM $Na_2HPO_4$, 25 mM citric acid, and 0.006% $H_2O_2$. After each incubation step, three washes were performed in Tris-buffered saline (TBS) buffer (20 mM Tris, 150 mM NaCl, pH 7.4) with 0.5% bovine serum albumin (BSA) and 0.05% Tween 20. For each target, we sequenced 36 clones with a signal at least 30% above the background. We included only clones with unambiguous and unique sequences for further characterization.

## Medium-scale expression of sybodies and characterization with size exclusion chromatography

*E. coli* MC1061 F- cells (Lucigen, Wisconsin, USA) with unique sybody clones were cultured in separate flasks in 50 mL of TB medium with 34 µg/mL chloramphenicol at 160 rpm at 37°C. On reaching an OD600 of 0.4–0.8, the temperature was reduced to 22°C, and the expression was induced with 0.02% (wt/vol) L-arabinose. After induction, the cells were cultured overnight at 160 rpm and 22°C. The periplasmic extracts were prepared using a sucrose osmotic shock, and the sybodies were purified with QIAGEN Ni-NTA resin in a TBS buffer. GroEL2 sybodies were further purified and analyzed by SEC using an SRT10C-100 column (Sepax Technologies, Delaware, USA). The ideal monomeric sybodies eluted at 11–12.5 mL. After purification with Ni-NTA resin, the GroEL1 sybodies were desalted using a PD Minitrap column and analyzed by analytical SEC with an SRT SEC-100 column (Sepax Technologies). With this column, the ideal monomeric sybodies eluted at 7–8 mL. Monomeric sybodies were chosen for further characterization.

## Affinity measurements with bio-layer interpherometry (BLI) using an Octet RED96 system

For the GroEL2 nanobodies, high-precision Streptavidin (SAX) probes were pre-equilibrated in an assay buffer: 50 mM Tris-HCl pH 7.5, 150 mM NaCl, 0.5% BSA, and 0.05% Tween 20. The baseline was measured for 180 s followed by 200 s of coating with biotinylated GroEL protein at 5 µg/mL. The probes were then incubated with clones of sybodies at 200 nM for 500 s followed by 800 s of dissociation time. The approximate affinities ($K_D$) were determined based on single concentration measurements of on and off rates. The measurements were carried out at 22°C. Data were reference-subtracted and aligned with each other in the Octet Data Analysis software v10.0 (FortéBio, California, USA) using a 1:1 binding model. For GroEL1, the protocol was the same except that penta-anti-His probes were used, and the sybodies were immobilized.

## Sybody-binding assays

### Western blot–based detection of sybody binding to cultured Mmr biofilms

One-week-old Mmr biofilm cultures (100 µL) were pelleted and washed once with 200 µL of TBS with 0.5% BSA (wt/vol) (TBS-BSA). The biofilms were incubated for 30 min at 28°C with 0.1, 1, or 5 µg of Myc-His-tagged GroEL1 sybodies or 1 or 5 µg of Myc-His-tagged GroEL2 sybodies in TBS-BSA or with 1 or 5 µg of MBP-sybody as a negative control. The pellets were washed three times with 200 µL of TBS-BSA and resuspended in 100 µL of TBS. Four-times concentrated protein-loading buffer (Licor, Nebraska, USA) without reducing agents (DTT or 2-mercaptoethanol) was added to the samples that were boiled for 20 min at +95°C. The heat-denatured proteins were separated in a 12% Tris-Glycine SDS-PAGE, transferred onto nitrocellulose membranes that were then blocked with TBS supplemented with 0.05% Tween (vol/vol) and 1% BSA (TBSTB) (wt/vol) overnight at +4°C. The blots were then incubated for 2 hours at RT with either a monoclonal mouse anti-His-tag antibody (Merck, Darmstadt, Germany) for GroEL1 sybody experiments or a monoclonal mouse anti-c-Myc antibody (Merck) for GroEL2 sybody experiments, using both antibodies at a dilution of 1:5,000 in TBSTB. The blots were washed three times with TBS-containing 0.2% Tween (vol/vol) (TBST), incubated for 1 hour at RT with 15,000-times diluted IRDye 800CW Donkey anti-Mouse IgG secondary antibody (Licor), washed again three times with TBST, and imaged using the Odyssey DLx fluorescence imager (Licor).

### Mmr infections of adult zebrafish and collecting granulomas

Mmr with tomato fluorescence pTEC27 plasmid was used in the granuloma experiments in zebrafish. pTEC27 was a gift from Lalita Ramakrishnan (Addgene plasmid no. 30182). The bacterial culturing and intraperitoneal injections were performed as described earlier (18) except 75 µg/mL of hygromycin was used as a selection marker for the

Mmr strain including pTEC27 plasmid. The infection dose was 75 cfu. Granulomas were collected at 8 wpi utilizing the red fluorescence signal from Mmr and a NightSea lamp with an emission filter of 600 nm (Electron Microscopy Science, Pennsylvania, USA) and stored at −80°C.

### Imaging sybody binding against GroEL in mycobacterial biofilms in vitro

One- and two-week-old Mmr and Mtb *in vitro* biofilms were used to study GroEL1 and GroEL2 sybody binding on intact biofilms. Biofilm mass was collected by centrifuging for 3 min at 2,000*g* and washed three times with PBS. The samples were then blocked with 2% BSA (Sigma-Aldrich) in PBS at RT for 30–60 min. The biofilms were incubated with myc-tagged sybody against GroEL1 or GroEL2 in 0.1% BSA in PBS at RT for 1.5 hours. After sybody incubation, the samples were washed twice with 2% BSA in PBS and incubated with 5 µg/mL of myc tag monoclonal antibody (myc.A7) Dylight 488 (Invitrogen) in 2% BSA in PBS at 4°C for 30–60 min. The antibody solution was removed, and the samples were washed twice with PBS and once with $H_2O$. The biofilm samples were transferred onto microscopy slides, and excess liquid was carefully removed before the samples were mounted with ProLong Diamond Antifade Mountant with DAPI (Invitrogen) and covered with a high-precision cover slide. The imaging was performed using a Nikon A1R+ confocal microscope with a 60× oil immersion objective.

### Imaging sybody binding against GroEL of ex vivo granulomas

Granuloma samples were first blocked with 2% BSA in PBS at room temperature for 1.5 hours and then stained with 100 µg/mL of nanobody in 0.1% BSA in PBS for 4 days at 4°C. To label the sybodies for granuloma staining, the sybodies were incubated with Alexa Fluor 488 NHS-label (Invitrogen) in 1:2 molar excess of the dye in PBS at room temperature for 1 hour with gentle agitation. Ten milligram per milliliter of glycine was added after the staining step to end the reaction, and unbound dye was removed with dialysis in PBS. After staining, the unbound sybody was removed by washing twice with PBS and once with $H_2O$. The samples were mounted with ProLong Diamond Antifade Mountant with DAPI (Invitrogen) and imaged using a Nikon A1R+ confocal microscope.

In all cases, a full z-stack was acquired with equal laser power and photomultiplier voltage across all samples. The maximum intensity projections of the acquired z-stacks were created and analyzed for fluorescence intensity comparisons.

## RESULTS

### Biotinylated proteins were successfully captured from the intact Mmr biofilms

The workflow illustrated in Fig. 1 outlines the steps used to capture the most accessible biofilm proteins produced by Mmr *in vitro* and *in vivo* and to generate sybodies that bind selected proteins within the mycobacterial biofilms both *in vitro* and in granulomas. First, proteins on intact Mmr biofilms were subjected to biotinylation with cell-impermeable sulfo-NHS-LC-biotin, streptavidin affinity purification, and LC-MS/MS to identify proteins accessible as targets for sybodies. The biotinylation of the biofilm proteins was conducted in conditions that keep Mmr cells intact (11) using non-biotinylated biofilms as negative controls. When membrane-impermeable sulfo-NHS-LC-biotin labeling is applied to intact biofilms, the biotin label becomes attached to the externally available free amino groups on the exposed proteins. The deeper the protein is localized, the less likely it is to become labeled. Hence, when pulled down with streptavidin beads, the collected protein pool is enriched with surface-exposed proteins. As negative controls, the non-biotinylated control samples were not treated with sulfo-NHS-LC-biotin but were lyzed similarly to the biotinylated sample and the proteins were pulled down with streptavidin beads in separate tubes. The streptavidin pull-down from the non-biotinylated samples shows the level of non-specific binding to the streptavidin beads/tube and the endogenously biotinylated proteins. Hence, if a certain protein

was also pulled down in the non-biotinylated sample to the same extent as in the surface-biotinylated sample type, it was removed from the hit list, as its presence in the sample was likely to be due to non-specific binding. Proteins that were more abundant in the pull-down were identified as likely surface-exposed proteins. Table S1 lists all the proteins identified from the biotinylated and non-biotinylated biofilms. In total, 3,080 and 1,257 proteins were detected in at least two out of the three biological replicate samples from the biotinylated and non-biotinylated biofilms, respectively (Table S1). Then, both identification data sets were compared using label-free quantification (LFQ), which indicated ca. three to five times higher intensity values ($P < 0.05$) for inherently biotinylated proteins, such as biotin synthase (catalyzing the conversion of dethiobiotin to biotin, BIOB_MYCMM) and biotin-dependent carboxylase (A0A2Z5Y992_MYCMR) after biotinylation, which confirms that adequate protein biotinylation efficiency was obtained under the conditions used (Table S2). In addition, the highest raw intensity values within non-biotinylated samples were obtained for one of the naturally biotinylated proteins, a biotin carboxyl carrier protein (B2HDZ3_MYCMM), further validating the streptavidin-based purification strategy used in this study. A principal component analysis (PCA) (PC1 explaining 71% of the total variance and PC2 12.4%) on the LFQ data indicated the biotinylated proteomes clustering closely together and being clearly separated from those associated with the non-biotinylated data. The heatmap shown in Fig. 2B indicates higher protein abundances for most of the matrix-associated proteins in comparison to their non-biotinylated counterparts, further confirming that the proteins exposed at the biofilm surface were successfully biotinylated, captured, and identified.

## One hundred sixty proteins are more abundant during biofilm than planktonic growth mode

Next, all identified proteins from the biotinylated and non-biotinylated identification data sets were quantitatively compared to indicate statistically significant protein abundance changes. Table S2 lists 448 proteins with significant abundance change (t-test, $P < 0.05$) and with predicted secretion motifs and/or subcellular location [TMHMM, isoelectric point (pI), SignalP 6.0]. Among these proteins, only 9 harbor a signal peptide (SPI or SPII type) directing the protein to the cell wall/exterior. Six of these are potential lipoproteins (presence of lipobox), while one is equipped with 12 transmembrane domains (TMDs) anchoring the protein to the mycomembrane. From all 448 proteins, 19 proteins have 2–14 TMDs, indicating their likely location in the mycomembrane. Majority of the identified proteins (n = 420) detected without TMDs or classical secretion motifs have been listed in the MoonProt database as proteins with moonlighting functions (31), suggesting that these proteins have entered the biofilm matrix via non-classical routes. Altogether, 432 proteins show ≥1.5 times higher abundance compared to their non-biotinylated counterparts (Table S2) and are considered potential surface-exposed targets on mycobacterial biofilms.

Then, we wished to investigate whether the identified proteins enriched on the biofilms (containing both the pellicle- and submerged-type biofilms) are more abundant during the biofilm mode of growth in comparison to planktonic growth. For this purpose, we used the LFQ identification data reported recently for the same Mmr strain grown in planktonic and biofilm forms (11). The LFQ data obtained by cell surface tryptic shaving included the identifications from both the planktonic cell surfaces after 4 days of growth and from pellicle- and submerged-type biofilms after 4 weeks of growth at +28°C. Table S3 lists 904 proteins with significantly higher abundance increase (unpaired t-test, $P < 0.05$) on both the pellicle- and/or submerged-type biofilms in comparison to the same proteins on the planktonic cell surfaces. The Venn diagram in Fig. 2C, comparing the number of these identifications with those obtained by the biotinylation proteomics in this study, indicated that ca. 20% of the proteins ($n = 158$) could be detected by both the tryptic in vitro–shaving proteomics and by the proteomic approach used in this study (Table S4). Thus, we suggest that these shared proteins are more abundant on the biofilm

# Therapy-recruiting nanobodies against biofilms

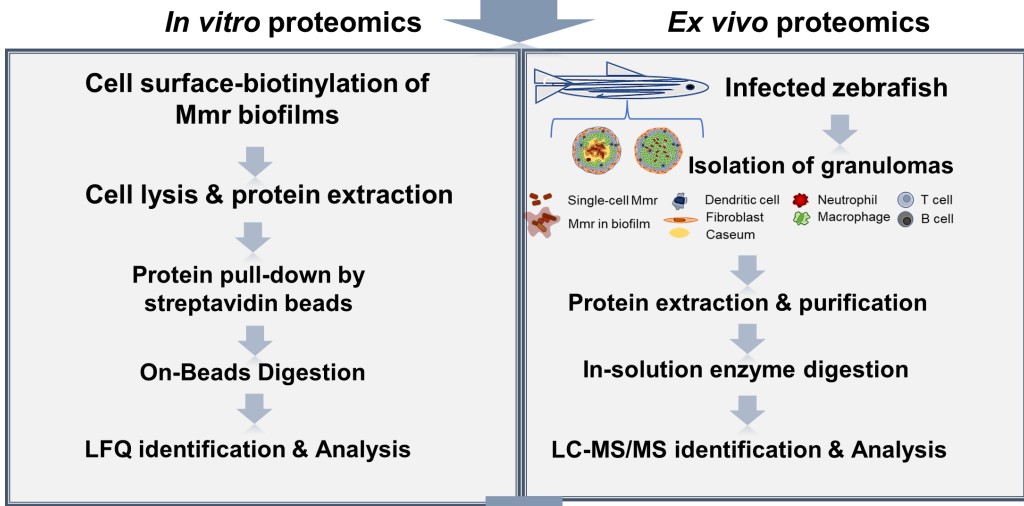

**In vitro proteomics**

**Cell surface-biotinylation of Mmr biofilms**

**Cell lysis & protein extraction**

**Protein pull-down by streptavidin beads**

**On-Beads Digestion**

**LFQ identification & Analysis**

**Ex vivo proteomics**

**Infected zebrafish**

**Isolation of granulomas**

Single-cell Mmr    Dendritic cell    Neutrophil    T cell
Mmr in biofilm    Fibroblast    Macrophage    B cell
Caseum

**Protein extraction & purification**

**In-solution enzyme digestion**

**LC-MS/MS identification & Analysis**

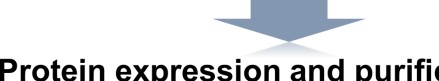

## Selection of target protein(s)

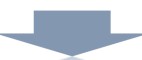

## Protein expression and purification

## Screening sybodies against selected protein targets

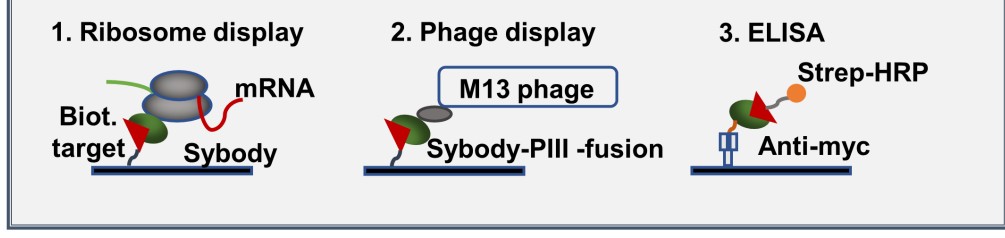

**1. Ribosome display**

mRNA

Biot. target    Sybody

**2. Phage display**

M13 phage

Sybody-PIII -fusion

**3. ELISA**

Strep-HRP

Anti-myc

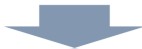

## Sybody expressions and characterizations

## Validation of sybody binding

**In vitro**

**Sybodies with intact Mmr biofilms**

**Immunoblotting confirms in vitro binding**

**Ex vivo**

**Sybodies with Mmr lesions dissected from zebrafish**

**CFM confirms ex vivo binding**

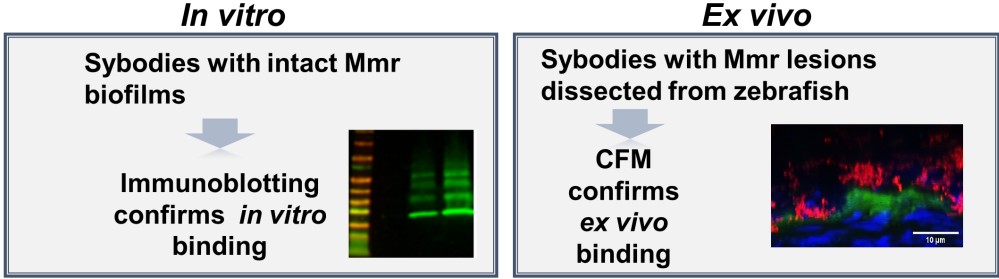

**FIG 1** The panel shows the general workflow of the study. The promising targets on the biofilm surface were identified by surface biotinylation of intact biofilms followed by lysis, protein extraction, and proteomic analysis. Mmr lesions, called

**FIG 1** (Continued)

granulomas, were collected from zebrafish, of which the total soluble proteome was determined to identify the surface hits also present in infected lesions. Two selected target proteins were purified and used as targets in a screen utilizing sybody libraries. Selected sybodies were subjected to binding tests on intact biofilms and analyzed by Western blotting and microscopy. CFM, confocal microscopy.

## GroEL paralogs were among the most abundant proteins *in vitro* and *ex vivo*

The top 20 proteins with significantly higher abundances on the biotinylated biofilms and with the highest raw intensity values in comparison to their non-biotinylated counterparts are shown in Fig. 5. All 20 proteins are predicted to have moonlighting functions, as indicated by the MoonProt 3.0 database (31). Computational predictions (SignalP 6.0) indicated that these proteins use a yet-unknown mechanism to reach the biofilm matrix. All top 20 proteins had at least one ortholog in Mtb. The molecular chaperone GroEL2 (A0A117DW44_9MYCO) was detected with the highest raw intensity values from the biofilm matrix. GroEL1 (B2HD10_MYCMM), a paralog of GroEL2, was also identified among the top eight proteins with high-intensity values. Both chaperones also displayed statistically significant abundance differences after biotinylation: GroEL1 with eight times and GroEL2 with four times higher LFQ values (t-test, $P < 0.05$) compared with those chaperones without biotinylation (Table S1).

To complement/confirm these proteomic results, we also performed *ex vivo* proteomics on granulomas dissected from Mmr-infected adult zebrafish at 8-week postinfection (wpi). Since the surface biotinylation proteomic strategy could not be performed on homogenized granulomas, likely due to free amines (released from tissues) inactivating the NHS-biotin (data not shown), we used a global proteomic approach instead. To this end, proteins were extracted from pooled granuloma samples (in total 10 replicate samples and each with 10 granulomas) under conditions preventing cell disruption as much as possible but promoting the solubilization of the mycobacterial proteins present either on the biofilm matrices or released into the granuloma environment. As there are no protocols to distinguish or remove planktonic bacteria from a complex biological sample, we cannot exclude the possibility that some of the detected mycobacterial proteins might have originated from planktonic cell surfaces. Table S5 lists 91 mycobacterial proteins that were detected in granulomas *ex vivo* with the aid of protein aggregation capture on microparticles (PACM) sample preparation and LC-MS/MS. Fig. 6 shows the top 30 mycobacterial proteins detected *ex vivo* with the highest raw intensity values. Again, GroEL2 was detected with the highest intensity values from each of the 10 replica granuloma samples, while its paralog, GroEL1, was identified from 9 out of the 10 replicates with lower raw intensity values. Fig. 2 indicates that 49 proteins could be commonly identified by biotinylation and *ex vivo* proteomics. The list of these proteins is presented in Table 1. Altogether, 33 proteins were shared by both *in vitro* biofilm surface proteomes obtained by the biotinylation and surface-shaving proteomics (Table S4). The data obtained on GroEL1 and GroEL2 within the *in vitro* biofilm proteome identified by the shaving proteomics indicate that the abundance of GroEL2 is increased by ca. four times (t-test, $P < 0.05$) on the 4-week-old biofilms, but GroEL1, while detected with reasonably high LFQ values on the biofilm matrices at this time point, was slightly less abundant on the biofilms in comparison to the planktonic cell surfaces (Table S3). However, GroEL1 was detected as ca. two times (t-test, $P < 0.05$) more abundant on the 2-day-old and 1-week-old biofilm matrices compared to this chaperone on the planktonic cells (Table S3), which is in line with the earlier report indicating that GroEL1 is needed for the biofilm formation in mycobacteria (32). Thus, these findings imply the suitability of both GroEL2 and GroEL1 as promising targets for sybody binding.

**TABLE 1** Condensed results from the sybody screenings against Mmr GroEL1 and GroEL2

| Sybody library against GroEL1 | Enrichment in phage display 1 | Enrichment phage display 2 | No. of ELISA hits 1.3-fold above negative control | No. of unique binders (total sequenced) | No. of well-behaved binders in SEC and BLI (total analyzed) | No. of binders to intact biofilms (total tested) |
|---|---|---|---|---|---|---|
| Concave | 2.2 | 1,109 | 11 (94) | 9 (12[a]) | 5 (9) | 0 (3) |
| Loop | 1.9 | 223 | 7 (94) | 4 (6[a]) | 3 (4) | 1 (2) |
| Convex | 6 | 5,288 | 18 (94) | 13 (18) | 3 (15) | 1 (2) |
| Sybody library against GroEL2 | Enrichment in phage display 1 | Enrichment in phage display 2 | No. of ELISA hits two fold above negative control | No. of unique binders (total sequenced) | No. of well-behaved binders in SEC and BLI (total analyzed) | No. of binders to intact biofilms (total tested) |
| Concave | 0.7 | 8.6 | 16 (94) | 3 (3) | 3 (3) | 1 (3) |
| Loop | 1.7 | 7.6 | 31 (94) | 26 (33) | 13 (15) | 3 (13) |

[a]One of the ELISA hits on the loop library turned out to be from the concave library.

## Selected sybodies produced against GroEL1 and GroEL2 bind intact *in vitro* biofilms

We selected GroEL2 as the primary target for testing the sybody binding since this chaperone was present both on the *in vitro* and *ex vivo* biofilms with the highest raw intensity values. We also generated sybodies against GroEL1 as this chaperone, although detected here with somewhat lower raw intensity values, contributes to the biofilm formation in mycobacteria (32) and is more abundantly produced during the first 2 weeks of biofilm growth. In addition, both have functional orthologs in Mtb, and the protein structure of Mtb orthologs produced in *Escherichia coli* has been previously determined (33, 34), demonstrating that the expression and purification of these chaperones in *E. coli* is feasible.

The recombinant GroEL1 and GroEL2 of Mmr were expressed and purified from *E. coli* in biotinylated and non-biotinylated forms. The purity, biotinylation, monodispersity, and stability of the proteins were assessed with SDS-PAGE, tamavidin shift assay, size exclusion chromatography (SEC), and nano differential scanning fluorimetry (nanoDSF), respectively (Fig. S1). The purified proteins were analyzed by mass spectrometry to verify their identity (Fig. S2) and were used as targets in the sybody screening platform developed in the Seeger Laboratory, as described in the literature (20, 21). Briefly, this platform allows the *in vitro* selection of binders from a randomized library of $10^{12}$ sybodies using ribosome display combined with two rounds of phage display. The main results of these screens are condensed in Table 1. We identified 26 and 29 unique binders against GroEL1 and GroEL2, respectively. Subsequent characterization included SEC and affinity measurements by bio-layer interferometry (BLI) on purified target proteins (results collated in Table 1). Based on the SEC data, only monomeric sybody clones with low-to-moderate column interactions were selected. Following the BLI measurements, sybody clones with a $K_D$ value below 1 µM were considered as hits. The approximate affinities to the purified target protein as determined in the BLI measurements can be found in Table 2. These criteria led to a final list of 7 and 16 GroEL1 and GroEL2 sybody clones, respectively.

We then tested the capacity for each clone to recognize respective epitopes on intact cultured Mmr biofilms. In the first instance, this involved the co-incubation of 1 or 5 µg of myc-His-tagged sybodies with intact Mmr biofilms. In this intact sample type, the sybody would only be able to bind if the target was surface-exposed. Unbound sybodies were washed off the biofilm pellets. The pellets with the bound sybodies were analyzed by SDS-PAGE, and the bound sybodies were detected by Western blot using Myc-tag and His-tag antibodies (Fig. 3 and Fig. S3A). Of the seven GroEL1-sybody clones tested, two were detectably binding to the biofilm surface. Of the 16 GroEL2 sybody clones, four showed detectable binding to mycobacterial biofilms. Some of the sybodies (GroEL1_SB31, GroEL2_SB3, GroEL2_SB9) formed additional bands on the gels

**TABLE 2** Characteristics of GroEL1 and GroEL2 sybodies

| Target | Sybody number | Molecular weight (Dalton) | Size exclusion chromatography profile | Approximate $K_D$ to purified target (nM) | Binding to Mmr biofilms | Protein sequence |
|---|---|---|---|---|---|---|
| GroEL1 | 1 | 15,767 | Monomeric | 130 | No | QVQLVESGGGLVQAGGSLRLSCAASGFPVEQRQMYWYR QAPGKEREW-VAAIQSYGKRTKYADSVKGRFTISRDNAKN TVYLQMNSLKPEDTAVYYCVVYVGG-GYKGQGTQVTVSA GRAGEQKLISEEDLNSAVDHHHHHH |
| GroEL1 | 5 | 15,648 | Monomeric | 127 | No | QVQLVESGGGLVQAGGSLRLSCAASGFPVEHKQMRWYR QAPGKEREWVAAIESSGQY-TIYADSVKGRFTISRDNAKNT VYLQMNSLKPEDTAVYYCFVGVGAGYYGQGTQVTVSAG RAGEQKLISEEDLNSAVDHHHHHH |
| GroEL1 | 9 | 15,522 | Monomeric | 64 | No | QVQLVESGGGLVQAGGSLRLSCAASGLPVWQQGMTWY RQAPGKEREWVAAIDSVGAQ-TYYADSVKGRFTISRDNAK NTVYLQMNSLKPEDTAVYYCAVNVGARYIGQGTQVTVSA GRAGEQKLISEEDLNSAVDHHHHHH |
| GroEL1 | 12 | 16,539 | Monomeric | 14 | NA[a] | QVQLVESGGGLVQAGGSLRLSCAASGFPVTQAWMEWY RQAPGKEREWVAAIFSHGGGT-FYADSVKGRFTISRDNAK NTVYLQMNSLKPEDTAVYYCNVKDTGERDNWYDYWGQ GTQVTVSAGRAGEQKLISEEDLNSAVDHHHHHH |
| **GroEL1** | **14** | **16,404** | **Mostly Monomeric/sticky** | **31** | **Yes** | **QVQLVESGGGLVQAGGSLRLSCAASGFPVXNAYMHWY RQAPGKEREWVAAILSSGAHT-LYADSVKGRFTISRDNAK NTVYLQMNSLKPEDTAVYYCNVKDYGAGVRYYDYWG QGTQVTVSAGRAGEQKLISEEDLNSAVDHHHHHH** |
| GroEL1 | 18 | 16,520 | Monomeric | 22 | No | QVQLVESGGGLVQAGGSLRLSCAASGFPVKTKHMYWYR QAPGKEREWVAAITSIGMI-TAYADSVKGRFTISRDNAKNT VYLQMNSLKPEDTAVYYCNVKDWGTNRQAYDYWGQGT QVTVSAGRAGEQKLISEEDLNSAVDHHHHHH |
| **GroEL1** | **31** | **17,198** | **Monomeric** | **92** | **Yes** | **QVQLVESGGGSVQAGGSLRLSCAASGTIYKIYYLGWFRQ APGKEREGVAALNTFSGGTYYADSVKGRFTVSLDNAKN TVYLQMNSLKPEDTALYY-CAAAYDMEGYAWPLYWWH YEYWGQGTQVTVSAGRAGEQKLISEEDLNSAVDHHHHH HH** |
| GroEL2 | 1 | 15,395 | Monomeric | 591 | No | QVQLVESGGGLVQAGGSLRLSCAASGFPVAITYMHWYR QAPGKEREWVAAISSTGKTT-WYADSVKGRFTISRDNAKN TVYLQMNSLKPEDTAVYYCLVEVGHYYKGQGTQVTVSAG RAGEQKLISEEDLNSAVDHHHHHHH* |
| GroEL2 | 2 | 15,087 | Monomeric | 157 | No | QVQLVESGGGLVQAGGSLRLSCAASGFPVSSSTMTWYR QAPGKEREWVAAIDSVGNE-TYYADSVKGRFTISRDNAKN TVYLQMNSLKPEDTAVYYCAVFVGSYYGQGTQVTVSAGR AGEQKLISEEDLNSAVDHHHHHH |
| **GroEL2** | **3** | **15,554** | **Monomeric** | **18** | **Yes** | **QVQLVESGGGLVQAGGSLRLSCAASGFPVAYWEMVWY RQAPGKEREWVAAIRSTGWKT-VYADSVKGRFTISRDNA KNTVYLQMNSLKPEDTAVYYCTAVYVGVHYKGQGTQV TVSAGRAGEQKLISEEDLNSAVDHHHHHH** |
| GroEL2 | 7 | 16,345 | Monomeric | 100 | No | QVQLVESGGGLVQAGGSLRLSCAASGFPVNDAWMYWY RQAPGKEREWVAAIMSMGFGT-WYADSVKGRFTISRDNA KNTVYLQMNSLKPEDTAVYYCNVKDRGKEHFSYDYWGQ GTQVTVSAGRAGEQKLISEEDLNSAVDHHHHHH |
| **GroEL2** | **8** | **16,234** | **Monomeric** | **13** | **Yes** | **QVQLVESGGGLVQAGGSLRLSCAASGFPVYMSWMYW YRQAPGKEREWVAAIMSEGAGT-WYADSVKGRFTISRD NAKNTVYLQMNSLKPEDTAVYYCNVKDTGSFHAQYDY WGQGTQVTVSAGRAGEQKLISEEDLNSAVDHHHHHH** |
| **GroEL2** | **9** | **16,333** | **Monomeric** | **277** | **Yes** | **QVQLVESGGGLVQAGGSLRLSCAASGFPVYQSWMYWY RQAPGKEREWVAAIMSDGSGT-WYADSVKGRFTISRDN AKNTVYLQMNSLKPEDTAVYYCNVKDFGHSRSRYDYW GQGTQVTVSAGRAGEQKLISEEDLNSAVDHHHHHH** |
| **GroEL2** | **10** | **16,343** | **Monomeric** | **193** | **Yes** | **QVQLVESGGGLVQAGGSLRLSCAASGFPVKHWYMHW YRQAPGKEREWVAAIQSTGSY-TAYADSVKGRFTISRDN AKNTVYLQMNSLKPEDTAVYYCNVKEYGFYHASYDYW GQGTQVTVSAGRAGEQKLISEEDLNSAVDHHHHHH** |
| GroEL2 | 11 | 16,259 | Monomeric | 271 | No | QVQLVESGGGLVQAGGSLRLSCAASGFPVDSAYMWWY RQAPGKEREWVAAIESNGEYT-FYADSVKGRFTISRDNAKN TVYLQMNSLKPEDTAVYYCNVKDTGAHHSYYDYWGQGT QVTVSAGRAGEQKLISEEDLNSAVDHHHHHH |
| GroEL2 | 16 | 16,232 | Monomeric | 15 | No | QVQLVESGGGLVQAGGSLRLSCAASGFPVSSSTMTWYR QAPGKEREWVAAIESWGAYT-WYADSVKGRFTISRDNAK NTVYLQMNSLKPEDTAVYYCNVKDYDGVADVIYDYWGQ GTQVTVSAGRAGEQKLISEEDLNSAVDHHHHHH |

*(Continued on next page)*

**TABLE 2** Characteristics of GroEL1 and GroEL2 sybodies (*Continued*)

| Target | Sybody number | Molecular weight (Dalton) | Size exclusion chromatography profile | Approximate $K_D$ to purified target (nM) | Binding to Mmr biofilms | Protein sequence |
|---|---|---|---|---|---|---|
| GroEL2 | 19 | 16,311 | Monomeric | 366 | No | QVQLVESGGGLVQAGGSLRLSCAASGFPVEWLEMAWYR QAPGKEREWVAAIYSYGME-TEYADSVKGRFTISRDNAKN TVYLQMNSLKPEDTAVYYCNVKDGGHAAWWYDYWGQ GTQVTVSAGRAGEQKLISEEDLNSAVDHHHHHH |
| GroEL2 | 22 | 16,211 | Monomeric | 507 | No | QVQLVESGGGLVQAGGSLRLSCAASGFPVYHSWMYWYR QAPGKEREWVAAIMSDGHGT-WYADSVKGRFTISRDNAK NTVYLQMNSLKPEDTAVYYCNVKDTGSSTTIYDYWGQGT QVTVSAGRAGEQKLISEEDLNSAVDHHHHHH |
| GroEL2 | 27 | 16,250 | Monomeric | 101 | No | QVQLVESGGGLVQAGGSLRLSCAASGFPVWKAYMWWY RQAPGKEREWVAAIESNGAYT-FYADSVKGRFTISRDNAK NTVYLQMNSLKPEDTAVYYCNVKDTGSDSENYDYWGQG TQVTVSAGRAGEQKLISEEDLNSAVDHHHHHH |
| GroEL2 | 28 | 16,134 | Monomeric | 35 | No | QVQLVESGGGLVQAGGSLRLSCAASGFPVDAYWMYWY RQAPGKEREWVAAIMSSGHGT-WYADSVKGRFTISRDNA KNTVYLQMNSLKPEDTAVYSCNVKDKGAQAAWYDYWG QGTQVTVSAGRAGEQKLISEEDLNSAVDHHHHHH |
| GroEL2 | 31 | 16,488 | Monomeric | 66 | No | QVQLVESGGGLVQAGGSLRLSCAASGFPVWMEWMYW YRQAPGKEREWVAAIMSEGDGT-WYADSVKGRFTISRDN AKNTVYLQMNSLKPEDTAVYYCNVKDFGYNNNYYDYWG QGTQVTVSAGRAGEQKLISEEDLNSAVDHHHHHH |
| GroEL2 | 32 | 16,078 | Monomeric | 153 | No | QVQLVESGGGLVQAGGSLRLSCAASGFPVVSQFMEWHR QAPGKEREWVAAIDSTGYST-FYADSVKGRFTISRDNAKNT VYLQMNSLKPEDTAVYYCNVKDAGEGQEQYDYWGQGT QVTVSAGRAGEQKLISEEDLNSAVDHHHHHH |
| GroEL2 | 33 | 16,356 | Monomeric | 244 | No | QVQLVESGGGLVQAGGSLRLSCAASGFPVYQHWMYWY RQAPGKEREWVAAIMSQGAGT-WYADSVRGRFTISRDNA KNTVYLQMNSLKPEDTAVYYCNVKDLGKAEYNYDYWGQ GTQVTVSAGRAGEQKLISEEDLNSAVDHHHHHH |

[a]NA, not applicable.

corresponding to larger proteins than the monomeric 15-kDa sybody. These bands are likely due to possible disulfide bonds mediating sybody–sybody interactions under the non-reducing experimental conditions used. The non-reducing sample buffer preserves existing disulfide bonds leaving multimeric complexes intact on the SDS-PAGE. Similar observations have been made with disulfide-containing milk whey proteins when analyzed without reducing agents (35).

## Confocal microscopy analysis confirms GroEL1 and GroEL2 sybody binding to *in vitro* biofilms

We next wanted to determine how the sybodies bind mycobacterial biofilms in a natural environment using confocal microscopy. We used Mmr and Mtb biofilms (cultured for 2 or 3 weeks) as targets for the myc-His-tagged GroEL1 and GroEL2 sybodies. We made use of two different staining strategies: one utilizing a fluorescently labeled anti-myc antibody binding to the myc-tagged sybodies on biofilms and another where sybodies were directly labeled with a green fluorophore. No signal was observed in the Mmr biofilms stained with the fluorescent anti-myc antibody alone (Fig. 4A), whereas the addition of the GroEL1 and GroEL2 sybodies induced staining of the biofilm (Fig. 4B and C). Furthermore, we observed low-intensity signals from 2-week-old avirulent Mtb biofilms (Fig. S3B and C). At 3 weeks, we used the fluorescently labeled sybodies (Fig. 4D through F) on cultured Mmr biofilms. At both time points and with both staining strategies, we could see sybodies binding to Mmr biofilms. The green fluorescence signal intensity acquired from maximum intensity projections was significantly different ($P < 0.001$) between the control and both the GroEL1 and GroEL2 sybody-stained images. GroEL2 sybody samples had a higher fluorescence intensity compared to GroEL1, which is in line with our binding assays (Fig. 3) and proteomic data (Fig. 5

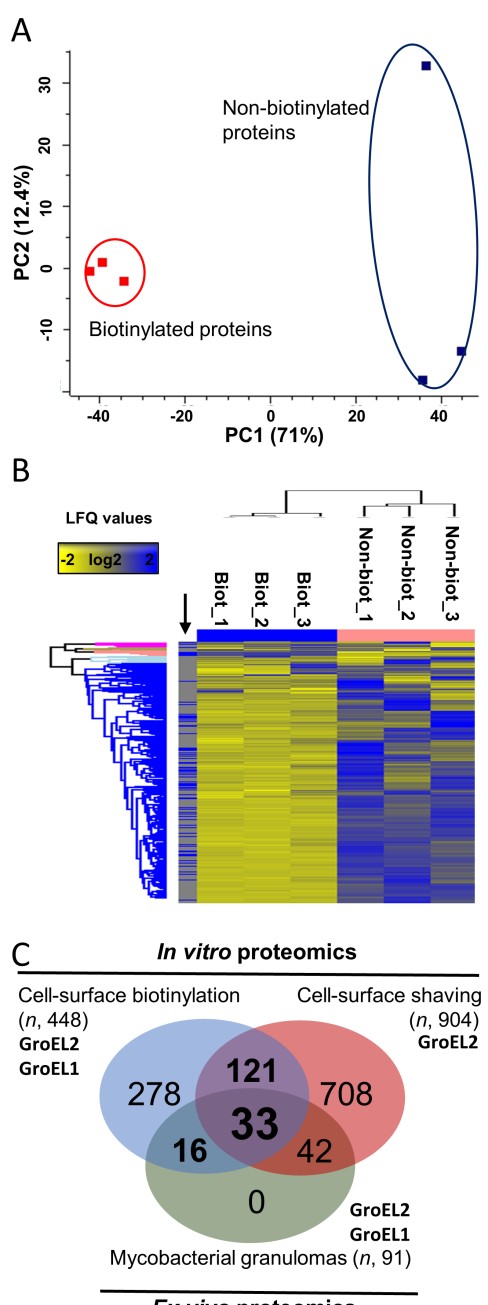

**FIG 2** Multivariate analyses on the non-biotinylated and biotinylated LFQ data, and Venn diagram comparing the number of Mmr proteins detected *in vitro* and *ex vivo*. (A) PCA plot based on the imputed and normalized LFQ data. Red and blue indicate the proteomes of three independent replica samples in the *in vitro* biotinylation experiment. (B) Hierarchical clustering of the biotinylated (Biot_1-3) and non-biotinylated (Non-biot_1-3) proteins (Distance = Euclidean; Linkage = Average) within each of the three biological replica samples. The black arrow refers to proteins significantly more abundant within the biotinylated protein samples (paired t-test and *P* < 0.05 with minimum two valid values in at least one of the groups). (C) Venn diagram comparing the number of biofilm matrix–associated proteins identified using *in vitro* and *ex vivo* proteomics approaches. Biotinylation proteomics with *n* = 448 corresponds to proteins showing statistically higher abundances compared to their non-biotinylated counterparts. Cell surface–shaving proteomics with *n* = 1034 corresponds to the number of the biofilm matrix proteins identified as significantly more abundant in comparison to their counterparts on planktonic cells (data from reference [10] ). *Ex vivo* proteomics with *n* = 91 refers to the number of mycobacterial proteins present in granulomas.

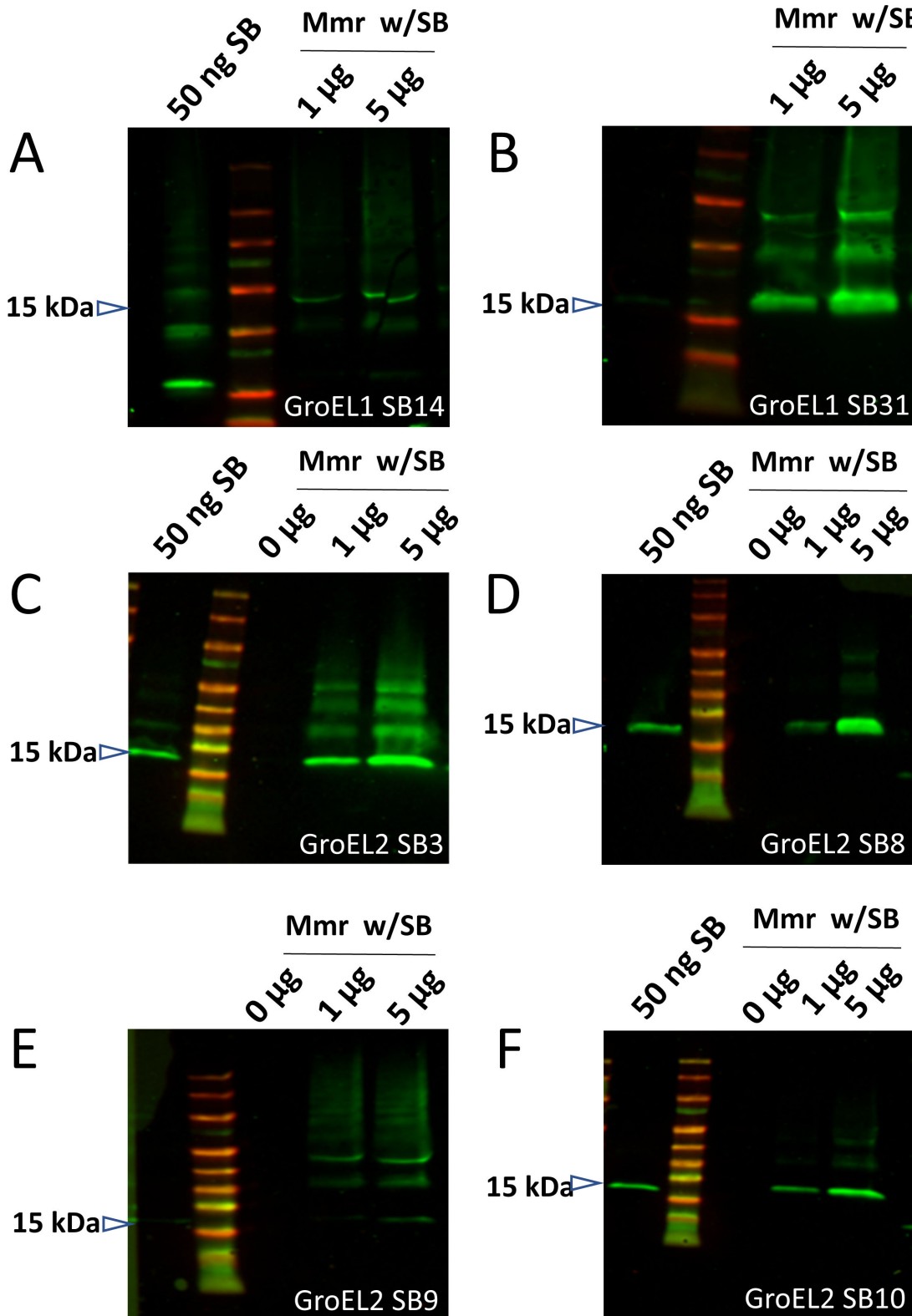

**FIG 3** Western blots detecting the sybodies (SB) bound onto the surface of Mmr biofilms after co-incubation with His-myc-tagged GroEL1 (A and B) and GroEL2 (C–F) sybodies. Mmr biofilms were incubated with 0.1, 1, or 5 µg of GroEL1 sybodies or 1 or 5 µg of His-GroEL2 sybodies followed by washes of the biofilm pellets, boiling in sample buffer, SDS-PAGE, and detection of the bound sybodies in the sample by Western blot using anti-His (in GroEL1-SB experiments) or anti-myc (in GroEL2-SB experiments) antibodies. In some of the experiments, the first lane (positive control) shows ~50 ng of the sybody directly loaded onto the gel. The negative control (0 µg) lane contains Mmr biofilm without sybodies.

and 6) showing GroEL2 to be a more abundant surface epitope on mycobacterial biofilms.

## GroEL1 and GroEL2 are surface-exposed epitopes on the biofilm inside granulomas

Subsequently, we aimed to assess whether the sybodies could also bind mycobacterial biofilms in granulomas isolated from zebrafish infected with red fluorescent Mmr. At 8 wpi, granulomas from the ovaries of female zebrafish were dissected based on their bright red fluorescence. The non-permeabilized granulomas were incubated with (green) fluorescent-labeled GroEL1 or GroEL2 sybodies and imaged with confocal microscopy. Fig. 4G through L show representative images of sybody binding into the biofilms in granulomas. In non-capsulated granulomas, the sybodies could stain the entire lesion (Fig. 4H and K; Video S2). In capsulated granulomas, without permeabilization and blood circulation, we could see the sybodies penetrate under the fibrous capsule and stain the adjacent areas of the biofilm (Fig. 4G and J; Video S1). These results show that we have identified GroEL-targeting sybodies that are not only able to bind biofilms in *in vitro* cultures but are also able to bind biofilms inside granulomas.

## DISCUSSION

Antibiotic-tolerant mycobacterial biofilms were recently shown to be present in TB (8). Such tolerance necessitates prolonged antibiotic treatment and potentially contributes to the development of antibiotic resistance (5). Thus, alternative biofilm-directed treatment modalities that do not rely solely on small molecule antibiotics open exciting horizons for more efficient treatment of TB.

Miniature single-domain antibodies, the so-called nanobodies that can be chemically and genetically linked to functional entities, could be used as a part of innovative research and treatment delivery strategies (9, 10). Unbiased omics approaches are useful for the identification of appropriate, abundant mycobacterial targets. Various existing high-quality proteomic data sets have described the total proteome of mycobacterial biofilms (36) and tuberculin (37) as well as a transcriptomic profile of biofilm-forming *Mycobacterium bovis* (38). An interesting first study from fixed human granuloma samples identified mycobacterial proteins specific to cellular versus caseous areas (39). Also, the surface proteome of planktonic forms of *Mycobacterium smegmatis* has been probed (40, 41). However, as nanobodies can only reach extracellularly exposed proteins, for the purpose of developing biofilm-binding nanobodies, there is a need to reliably identify abundant surface epitopes present on mycobacterial biofilms. Our laboratory previously published the first time-series experiment characterizing the dynamics of the mycobacterial biofilm matrix proteome (11). In the current study, we took a parallel approach to characterize the surface proteome of mature mycobacterial biofilm using surface biotinylation and streptavidin purification followed by mass spectrometry. We then compared these data with our pre-existing surface-shaving data from planktonic and biofilm Mmr cultures to reliably identify proteins that are specifically enriched on the biofilm surface. Subsequently, we extracted granulomas from Mmr-infected adult zebrafish and studied the mycobacterial proteome at late-stage infection to identify biofilm surface proteins present during infection. Combining the data from the three experimental setups allowed us to identify 33 proteins that are more abundant on the biofilm surface than on planktonic cells and are also highly expressed during the late-stage infection.

Cell-impermeable NHS-biotin-based strategies are a commonly used and gentle way of purifying surface-exposed proteins from bacteria (42). This strategy was previously successfully used for studying the surface proteome of planktonic mycobacteria (40, 41). Biotinylation strategies based on chemicals such as the sulfo-NHS-LC-biotin used here are essentially cell impermeable. However, when used in bacteria with a peptidoglycan layer, it is known that they can penetrate the peptidoglycan to some extent, and hence

## 2-week-old *M.marinum* biofilm cultures

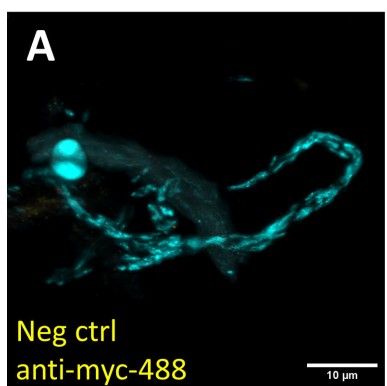
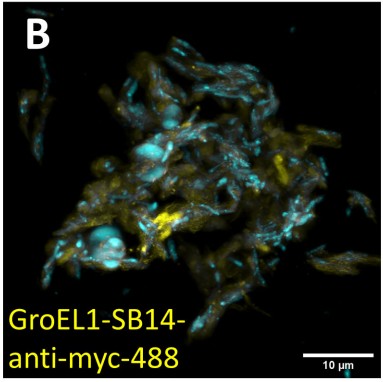
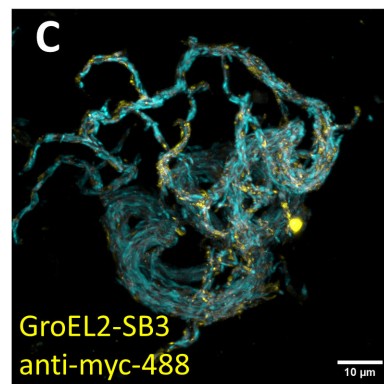

## 3-week-old *M.marinum* biofilm cultures

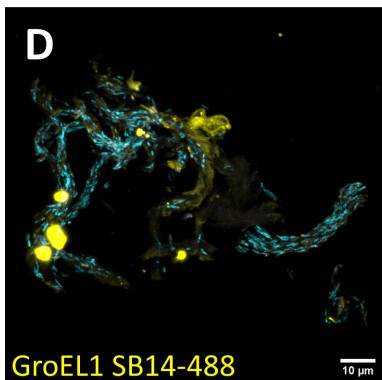
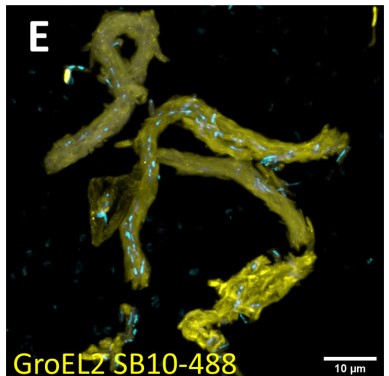
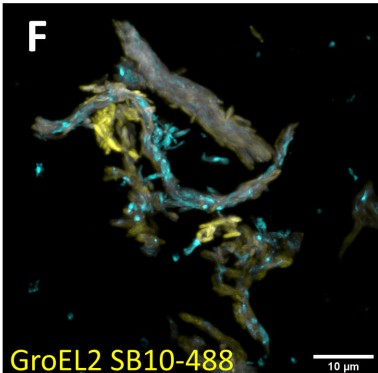

## Granulomas at 8 weeks post infection

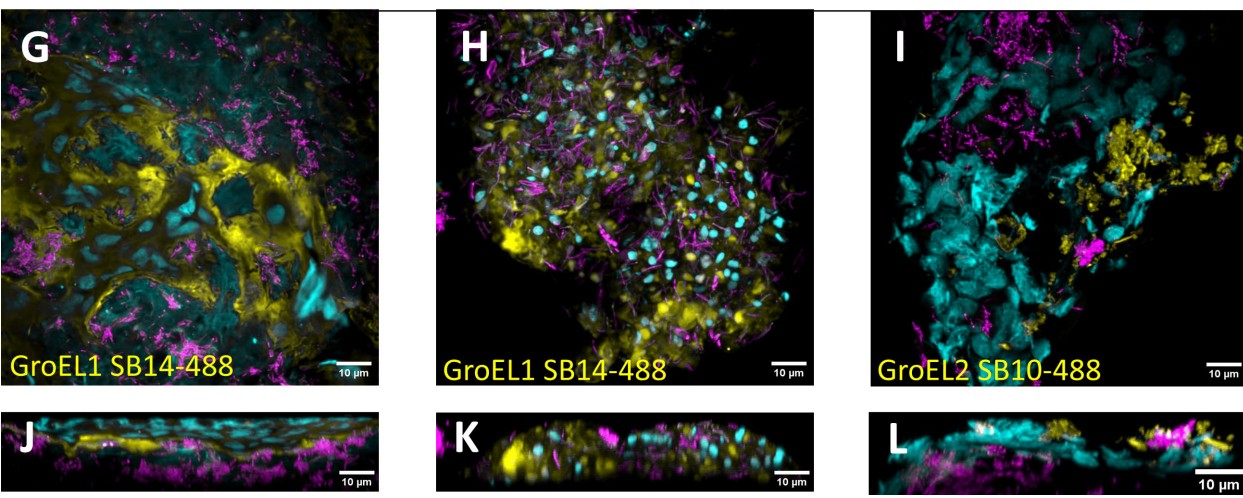

**FIG 4** GroEL1 and GroEL2 sybodies bind biofilms *in vitro* as well as Mmr inside granulomas *ex vivo* in microscopy experiments. DAPI (cyan) was used as a counterstain for DNA. Mycobacterial biofilms were stained with green fluorescence-labeled sybodies (yellow) (D–F) or with myc-tagged sybodies + green fluorescent anti-myc antibodies (yellow) (B and C) or green fluorescent anti-myc antibody only as a negative control (yellow) (A). In the granuloma experiments (G–I front view, J–L side view), bacteria (purple) expressing a red fluorescent protein were used to allow the extraction of the granulomas. Granulomas were incubated with 100 µg/mL of green fluorescence-labeled GroEL-binding sybodies and imaged with confocal microscopy. All scale bars are 10 µm.

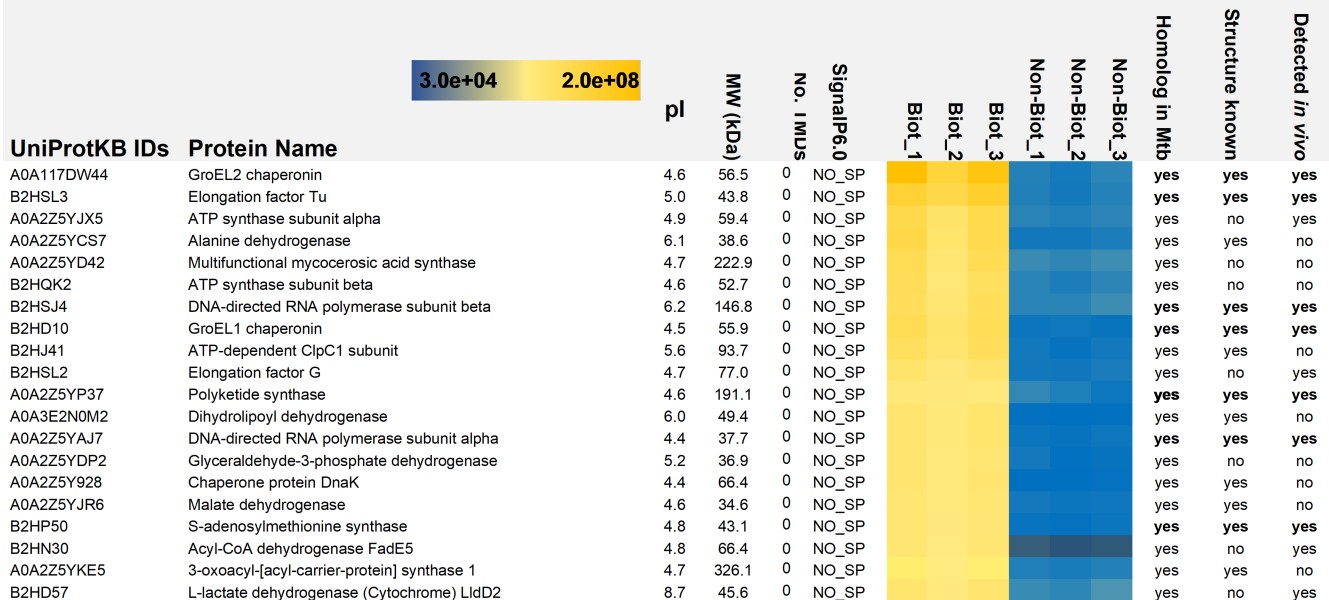

| UniProtKB IDs | Protein Name | | pI | MW (kDa) | NO. I MUs | SignalP6.0 | Biot_1 | Biot_2 | Biot_3 | Non-Biot_1 | Non-Biot_2 | Non-Biot_3 | Homolog in Mtb | Structure known | Detected *in vivo* |
|---|---|---|---|---|---|---|---|---|---|---|---|---|---|---|---|
| A0A117DW44 | GroEL2 chaperonin | | 4.6 | 56.5 | 0 | NO_SP | | | | | | | **yes** | **yes** | **yes** |
| B2HSL3 | Elongation factor Tu | | 5.0 | 43.8 | 0 | NO_SP | | | | | | | yes | yes | yes |
| A0A2Z5YJX5 | ATP synthase subunit alpha | | 4.9 | 59.4 | 0 | NO_SP | | | | | | | yes | no | yes |
| A0A2Z5YCS7 | Alanine dehydrogenase | | 6.1 | 38.6 | 0 | NO_SP | | | | | | | yes | yes | no |
| A0A2Z5YD42 | Multifunctional mycocerosic acid synthase | | 4.7 | 222.9 | 0 | NO_SP | | | | | | | yes | no | no |
| B2HQK2 | ATP synthase subunit beta | | 4.6 | 52.7 | 0 | NO_SP | | | | | | | yes | no | no |
| B2HSJ4 | DNA-directed RNA polymerase subunit beta | | 6.2 | 146.8 | 0 | NO_SP | | | | | | | **yes** | **yes** | **yes** |
| B2HD10 | GroEL1 chaperonin | | 4.5 | 55.9 | 0 | NO_SP | | | | | | | **yes** | **yes** | **yes** |
| B2HJ41 | ATP-dependent ClpC1 subunit | | 5.6 | 93.7 | 0 | NO_SP | | | | | | | yes | yes | no |
| B2HSL2 | Elongation factor G | | 4.7 | 77.0 | 0 | NO_SP | | | | | | | yes | no | yes |
| A0A2Z5YP37 | Polyketide synthase | | 4.6 | 191.1 | 0 | NO_SP | | | | | | | **yes** | **yes** | **yes** |
| A0A3E2N0M2 | Dihydrolipoyl dehydrogenase | | 6.0 | 49.4 | 0 | NO_SP | | | | | | | yes | yes | no |
| A0A2Z5YAJ7 | DNA-directed RNA polymerase subunit alpha | | 4.4 | 37.7 | 0 | NO_SP | | | | | | | **yes** | **yes** | **yes** |
| A0A2Z5YDP2 | Glyceraldehyde-3-phosphate dehydrogenase | | 5.2 | 36.9 | 0 | NO_SP | | | | | | | yes | no | no |
| A0A2Z5Y928 | Chaperone protein DnaK | | 4.4 | 66.4 | 0 | NO_SP | | | | | | | yes | yes | no |
| A0A2Z5YJR6 | Malate dehydrogenase | | 4.6 | 34.6 | 0 | NO_SP | | | | | | | yes | yes | no |
| B2HP50 | S-adenosylmethionine synthase | | 4.8 | 43.1 | 0 | NO_SP | | | | | | | **yes** | **yes** | **yes** |
| B2HN30 | Acyl-CoA dehydrogenase FadE5 | | 4.8 | 66.4 | 0 | NO_SP | | | | | | | yes | no | yes |
| A0A2Z5YKE5 | 3-oxoacyl-[acyl-carrier-protein] synthase 1 | | 4.7 | 326.1 | 0 | NO_SP | | | | | | | yes | yes | no |
| B2HD57 | L-lactate dehydrogenase (Cytochrome) LldD2 | | 8.7 | 45.6 | 0 | NO_SP | | | | | | | yes | no | yes |

**FIG 5** List of intact biofilm matrix proteins (Biot_1-3) identified with the highest raw intensity values. NO_SP protein carries no recognizable signal peptide and is predicted to enter the biofilm matrix via non-classical and yet-unknown pathway. Non-Bio_1-3 refer to proteins identified from the intact biofilms without biotinylation. Color gradient indicates the high (yellow) and low (blue) identification raw intensity values for the indicated proteins.

**RAW INTENSITY VALUES**

| UniprotKB ID | Protein name | Repl_1 | Repl_2 | Repl_3 | Repl_4 | Repl_5 | Repl_6 | Repl_7 | Repl_8 | Repl_9 | Repl_10 |
|---|---|---|---|---|---|---|---|---|---|---|---|
| **A0A117DW44** | **GroEL2 chaperonin** | | | | | | | | | | |
| B2HSL3 | Elongation factor Tu | | | | | | | | | | |
| A0A2Z5YCH3 | Phthiocerol dimycocerate exporter MmpL7 | | | | | | | | | | |
| A0A2Z5YEF1 | Integration ht factor | | | | | | | | | | |
| B2HQM8 | Two-component sensor and regulator | | | | | | | | | | |
| A0A3E2MWQ5 | Potential acyltransferase | | | | | | | | | | |
| B2HN62 | Holliday junction ATP-dependent DNA helicase RuvA | | | | | | | | | | |
| B2HJ81 | Cold shock protein A CspA_1 | | | | | | | | | | |
| B2HD09 | 10 kDa chaperonin | | | | | | | | | | |
| A0A100IF16 | Nucleoid-associated protein Lsr2 | | | | | | | | | | |
| A0A117DYA5 | ESAT-6-like protein | | | | | | | | | | |
| B2HR04 | Iron-regulated conserved protein | | | | | | | | | | |
| B2HHR5 | Acyl carrier protein | | | | | | | | | | |
| A0A2Z5YA35 | 50S ribomal protein L7/L12 | | | | | | | | | | |
| A0A2Z5YFU6 | Catalase-peroxidase | | | | | | | | | | |
| B2HGQ8 | Antigen 84 | | | | | | | | | | |
| B2HII3 | DNA-binding protein HU homolog | | | | | | | | | | |
| A0A2Z5YBD3 | 2-methylcitrate dehydratase | | | | | | | | | | |
| A0A3E2MR45 | ATP synthase subunit alpha | | | | | | | | | | |
| A0A2Z5Y9K5 | Heparin-binding hemagglutinin | | | | | | | | | | |
| A0A100IC61 | Glyceraldehyde-3-phphate dehydrogenase | | | | | | | | | | |
| A0A2Z5YPB5 | ESAT-6-like protein | | | | | | | | | | |
| A0A100I595 | Peptidyl-prolyl cis-trans isomerase | | | | | | | | | | |
| A0A117DZM6 | 30S ribomal protein S2 | | | | | | | | | | |
| A0A100I3X5 | ATP synthase subunit beta | | | | | | | | | | |
| B2HD57 | L-lactate dehydrogenase (Cytochrome) LldD2 | | | | | | | | | | |
| B2HFB8 | 3-hydroxyacyl-CoA dehydrogenase | | | | | | | | | | |
| B2HSI8 | 50S ribomal protein L10 | | | | | | | | | | |
| B2HIE1 | Electron transfer flavoprotein (Alpha-subunit) FixB | | | | | | | | | | |
| A0A100I1L1 | GroEL1 chaperonin | | | | | | | | | | |

**FIG 6** List of the Mmr proteins identified with the highest raw intensity values from the mycobacterial granulomas from zebrafish at 8 wpi. Repl_1-10, 10 replica samples with proteins extracted from 10 granulomas in each. Color gradient indicates the high (yellow) and low (blue) identification raw intensity values for the indicated proteins.

protein epitopes embedded within the cell wall may also be exposed to biotinylation (42). Therefore, follow-up experiments with selected sybodies and intact biofilms are essential for the final verification of the surface availability of the epitopes.

Here, surface proteomics revealed cytoplasmic proteins as the main component of the mycobacterial extracellular proteome. This is in line with a number of previous studies reporting cell surface proteomes of different bacterial species grown either as planktonic or biofilm states (11, 42–57). Detection of an overwhelming number of cytoplasmic proteins, including the r-proteins, can be explained by the protein identification method used, which favors the identification of cell surface proteins that can be easily assessed by biotinylation and streptavidin capture–based technique. This also explains why the detection of structural proteins remained either below the detection limit or were identified with low-intensity values. These proteins frequently contain highly hydrophobic and complex regions, which often are difficult to identify due to the inherent lack of trypsin cleavage sites within these regions and to the tendency of the hydrophobic peptides to aggregate without solubilizing detergents (58). Growing evidence from different bacterial biofilms has also linked the presence of fibrous proteins, including amyloids or amyloid-like fibers, to a functional/structural role within the biofilm matrix; amyloids have been reported to serve as building blocks and provide mechanical robustness to the biofilm (59, 60). Such proteins have highly ordered beta-sheet-rich filamentous morphology, an ability to interact erroneously and generate insoluble/protease-resistant aggregates/fibrils (61), thereby making these proteins impossible to identify using the proteomic method and conditions used. This is because the identification of such protein structures requires the use of trifluoroacetic acid (TFA) or formic acid (FA) to disperse/solubilize the aggregates/fibrils prior to tryptic digestion and LC-MS/MS (62). Since the proteomic identification method used in this study requires capturing the target proteins in their native form, the use of these solvents during biotinylation would have affected the protein structure. Moreover, TFA or FA are typically used for inactive tryptic digestions, and therefore their application prior to enzymatic treatment is not possible. Thus, instead of or in addition to structural proteins, bacteria may also use exported/released ribosomal proteins (r-proteins) as structural proteins to stabilize and strengthen the biofilm integrity, as demonstrated with *Staphylococcus aureus* biofilms (44). r-proteins typically form the most dominant protein group at the bacterial cell surfaces (45–49, 51, 57, 58), which was also demonstrated in this study.

Since the identified cytoplasmic proteins do not possess a common mechanism driving their export out of the cells, the widespread opinion is that these proteins are released via a regulated/programmed process as part of the bacterial life cycle, which may involve autolysins, phenol-soluble modulins, phages and/or membrane vesicles (11, 50, 54–56, 63). In addition to their primary function within the bacterial cell, many of these cytoplasmic proteins show a secondary, i.e., moonlighting, function after being exported out of the cells ((31, 44, 64). GroEL is an example of such a protein that after being released out of the cells shows pH-dependent adherence to the biofilm matrix in *S. aureus* (34). This protein is highly expressed in response to stress to perform its primary chaperone function within the cell, but when present at the cell surface, it can act as an adhesive protein contributing to virulence (31, 43, 63, 64). Although our study provides proteome-wide information on the mycobacterial biofilms and confirms the presence of many cytoplasmic moonlighters on the biofilm matrix, detailed mechanisms underlying protein release onto the biofilm surface were not at the core of this study.

Here, chaperone proteins GroEL1 and GroEL2 were both on the list of the top 10 most abundant proteins among the identified biofilm surface proteins. GroEL proteins have been detected within the extracellular fractions of mycobacteria, with an important role in virulence (11, 65, 66) and in biofilm formation (32). In Mtb, GroEL2 was identified as one of the most abundant antigenic proteins on the pellicle-type biofilm cells (67),

further supporting its extracellular localization also during infection. This knowledge, along with the fact that there were existing functional protocols for producing these proteins in *E. coli* (33, 34), made GroEL1 and GroEL2 attractive candidates as the first sybody-binding targets.

We characterized dozens of unique sybody clones against GroEL1 and GroEL2 by SEC and BLI. Based on these measurements, we selected only monomeric sybodies with low carbohydrate–binding capacities and with high affinity to the recombinant target protein ($K_D$ < 1 µM) for further testing in biofilms. We found 7 GroEL1 and 16 GroEL2 sybodies fulfilling these criteria. Although all the selected sybodies had high affinity to their cognate proteins, only 25% of the sybodies were able to bind to their targets in the context of the biofilm. This shows us that *in vitro*-determined affinity approximation against the purified target alone is not sufficient for estimating the *in-situ* binding. This result is expected as in the complex environment of the biofilm, not all epitopes are present/exposed and freely available for binding. Some parts of the target are likely to interact with other components of the matrix, rendering them unavailable for sybody binding. Despite these steric hindrances, we succeeded in finding two and four sybody clones developed against GroEL1 and GroEL2, respectively, that bind biofilms in the natural context.

In conclusion, this study provides evidence that biofilm surface proteomics can act as a surrogate to identify surface-exposed epitopes on mycobacterial biofilms. We also show the first data assessing the most abundant mycobacterial proteins in biofilms extracted from Mmr granulomas. The identified proteins can be targeted with nanobodies. Since nanobodies are small and easily functionalized by chemical or genetic linkages, they constitute uniquely applicable tools for innovative clinical research and therapeutic strategies (68, 69). In this study, we developed GroEL1- and GroEL2-binding sybodies that were successfully used for the delivery of fluorophores to intact biofilms *in vitro* and *ex vivo*. The platform described here can aid the development of biofilm-targeting research and therapeutic strategies against mycobacterial and other biofilms.

## ACKNOWLEDGMENTS

We want to thank Jacob Scheurich, Julia Flock, and Arne Börgel for the practical support at the Protein Expression and Purification Core Facility, EMBL Heidelberg; Stephan Niebling at the Sample Preparation and Characterisation Core Facility at EMBL Hamburg, for the service of carrying out BLI measurements for GroEL2 sybodies; Leslie Pan for help in the setup of the sybody screening platform; Teemu Ihalainen for his advice on planning the microscopy experiments; Hannaleena Piippo for technical assistance in the experiments and laboratory organization; Joel Selkrig and Anna Sueki for discussions on bacterial surface proteomics; and Kerstin Putzker and Peter Sehr for their kind assistance with the equipment at the Chemical Biology Core Facility.

The following core facilities are acknowledged for their services and support, which were essential for this work: Protein Expression and Purification Core Facility, EMBL Heidelberg, Proteomics Core Facility at Oslo University Hospital, Zebrafish Core Facility, Tampere University, Proteomics Core Facility EMBL Heidelberg, Sample Preparation and Characterisation Core Facility at EMBL Hamburg, Tampere Imaging Facility.

The study was funded by the Academy of Finland: Postdoctoral Fellowship, 338624 (M.M.H.), Clinical Researcher funding 326674 (M.P.), Project funding 322010 (M.P.), Profiling funding 326584 (M.P.) and Project funding 348968 (M.P.); Tampere Tuberculosis Foundation (M.M.H., M.P., H.M.); Jane and Aatos Erkko Foundation (M.P.), Sigrid Jusélius Foundation (M.P.), Biocenter Finland (M.P.), Core Facilities programme of the South-Eastern Norway Regional Health Authority (T.A.N.), and Research Council of Norway INFRASTRUKTUR-programme (295910).

M.M.H.: study design and conceptualization, project coordination, funding acquisition, supervision, protein production and purification, *in vitro* proteomics sample preparation, sybody screening, sybody binding to biofilms method development,

manuscript writing. H.L.: *in vitro* proteomics sample preparation, sample collection, sybody binding tests to biofilms and microscopy, image analysis, manuscript writing. A.S.: sample collection, *in vitro* proteomics sample preparation, binding tests to *in vitro* biofilms, binding tests to granulomas. K.R.: construct design and supervision of protein production and purification. K.L.: biophysical characterization of target protein, sybody screening. T.C.: sybody screening. C.L.: supervision of sybody screening. H.M.: sample collection, *ex vivo* proteomics sample preparation. T.M.: microscopy and image analysis. M.A.S.: conceptualization, supervision on the use of sybody libraries. J.R.: proteomics experiments. T.N.: proteomics supervision, funding acquisition. K.S.: study design and conceptualization, *ex vivo* proteomics sample preparation, proteomics data analysis, manuscript writing, visualization. M.P.: study design and conceptualization, supervision, funding acquisition.

The corresponding author confirms on behalf of all authors that there have been no involvements that might raise the question of bias in the work reported or in the conclusions, implications, or opinions stated.

## AUTHOR AFFILIATIONS

[1]Faculty of Medicine and Health Technology, Tampere University, Tampere, Finland
[2]European Molecular Biology Laboratory, Heidelberg, Germany
[3]Centre for Structural Systems Biology, Hamburg, Germany
[4]Deutsches Elektronen-Synchrotron (DESY), Hamburg, Germany
[5]European Molecular Biology Laboratory, Hamburg, Germany
[6]Institute for Medical Microbiology, University of Zurich, Zurich, Switzerland
[7]Department of Immunology, University of Oslo, Oslo, Norway
[8]Oslo University Hospital, Oslo, Norway
[9]Faculty of Agriculture and Forestry, University of Helsinki, Helsinki, Finland

## AUTHOR ORCIDs

Milka Marjut Hammarén  http://orcid.org/0000-0001-9076-8782
Hanna Luukinen  http://orcid.org/0000-0002-9334-6304
Alina Sillanpää  http://orcid.org/0000-0003-4554-6595
Kim Remans  http://orcid.org/0000-0002-4394-7953
Karine Lapouge  http://orcid.org/0000-0003-0620-9553
Tânia Custódio  http://orcid.org/0000-0002-5723-9093
Christian Löw  http://orcid.org/0000-0003-0764-7483
Henna Myllymäki  http://orcid.org/0000-0002-6936-4879
Toni Montonen  http://orcid.org/0000-0003-3784-0890
Markus Seeger  http://orcid.org/0000-0003-1761-8571
Tuula A. Nyman  http://orcid.org/0000-0001-8787-5886
Kirsi Savijoki  http://orcid.org/0000-0001-5325-925X
Mataleena Parikka  http://orcid.org/0000-0001-5555-3815

## FUNDING

| Funder | Grant(s) | Author(s) |
| --- | --- | --- |
| Academy of Finland (AKA) | 338624 | Milka Marjut Hammarén |
| Academy of Finland (AKA) | 326674, 322010, 326584, 348968 | Mataleena Parikka |
| Tampereen Tuberkuloosisäätiö (Tampere Tuberculosis Foundation) | | Milka Marjut Hammarén |

| Funder | Grant(s) | Author(s) |
|---|---|---|
| Tampereen Tuberkuloosisäätiö (Tampere Tuberculosis Foundation) | | Mataleena Parikka |
| Jane and Aatos Erkko Foundation | | Mataleena Parikka |
| Sigrid Juséliuksen Säätiö (Sigrid Jusélius Stiftelse) | | Mataleena Parikka |
| Biocenter Finland (BF) | | Mataleena Parikka |
| Core Facilities programme of the South-Eastern Norway Regional Health Authority | | Tuula A. Nyman |
| Research Council of Norway INFRASTRUKTUR-programme | 295910 | Tuula A. Nyman |
| Tampereen Tuberkuloosisäätiö (Tampere Tuberculosis Foundation) | | Henna Myllymäki |

## AUTHOR CONTRIBUTIONS

Milka Marjut Hammarén, Conceptualization, Funding acquisition, Investigation, Methodology, Project administration, Supervision, Writing – original draft, Writing – review and editing | Hanna Luukinen, Formal analysis, Investigation, Methodology, Validation, Visualization, Writing – original draft, Writing – review and editing | Alina Sillanpää, Formal analysis, Investigation, Methodology, Validation, Writing – review and editing | Kim Remans, Methodology, Resources, Supervision, Writing – review and editing | Karine Lapouge, Investigation, Methodology, Writing – review and editing | Tânia Custódio, Formal analysis, Investigation, Methodology, Writing – review and editing | Christian Löw, Resources, Supervision, Writing – review and editing | Henna Myllymäki, Formal analysis, Investigation, Methodology, Writing – review and editing | Toni Montonen, Formal analysis, Investigation, Methodology, Visualization, Writing – review and editing | Markus Seeger, Conceptualization, Supervision, Writing – review and editing | Joseph Robertson, Data curation, Investigation, Methodology, Writing – review and editing | Tuula A. Nyman, Funding acquisition, Supervision | Kirsi Savijoki, Conceptualization, Formal analysis, Investigation, Methodology, Visualization, Writing – original draft, Writing – review and editing | Mataleena Parikka, Conceptualization, Funding acquisition, Supervision, Writing – review and editing

## DATA AVAILABILITY STATEMENT

The proteomics data sets generated and/or analyzed during the current study are available in the PRIDE repository with data set identifiers PXD033425 and PXD039416 for the *in vitro* and the *ex vivo* data, respectively. The other data sets generated during and/or analyzed during the current study and not included in the manuscript are available from the corresponding author on reasonable request.

## ETHICS APPROVAL

All experiments have been accepted by the Animal Experiment Board in Finland (under the Regional State Administrative Agency for Southern Finland) and were carried out in accordance with the EU-directive 2010/63/EU on the protection of animals used for scientific purposes and with the Finnish Act on Animal Experimentation Act 497/2013 and the Government Decree 564/2013. The use of genetically modified micro-organisms is done following the Finnish Genetic Engineering Law 377/1995, under the permission of Valvira.

## ADDITIONAL FILES

The following material is available online.

## Supplemental Material

**Table S1 (293065_1_supp_6572210_rph7my.pdf).** All identified proteins captured by streptavidin from the biotinylated and non-biotinylated intact and lysed biofilms. Secretory/subcellular location for each protein was predicted using EMBOSS Pepstats (pI, Mw, protein length), SignalP 6.0 (Moonlighters, T7SS, TAT, SecI-III), TMHMM (no. transmembrane domains). Protein sequences for each protein were retrieved from UniProt (https://www.uniprot.org/uploadlists/) with the aid of the UniProt IDs. IDs regarding GroEL1 and GroEL2 are indicated with bold letters and shaded in red. Cells in raw intensity value-associated columns shaded in grey indicate proteins that were not detected or were detected in only one of the replicate samples.

**Table S2 (293065_1_supp_6572212_rph7my.pdf).** List of proteins (n, 448) with significant higher abundancies after biotinylation (cells in intense red) and with predicted molecular weights, pI values, sequence lengths, and secretory motifs/subcellular location. Naturally biotinylated example proteins are in bold letter and shaded in grey.

**Table S3 (293065_1_supp_6572213_rpg7my.pdf).** List of proteins (n, 904) with significantly higher abundancies (unpaired T-test and $p < 0.05$, cells in deep blue) on biofilm cells in comparison to planktonic cells. Cells shaded in red indicate proteins showing signficantly increased abundance on the 4-week-old biofilm matrices in comparison to their equivalent proteins on the planktonic cell surfaces. Cells shaded in yellow show the LFQ values for GroEL1 on 2-day-, 1-week- and 2-week-old biofilm matrices with T-test statistics for pairwise comparisons. log, planktonic cells; B, submerged-type biofilms; P, pellicle-type biofilms.

**Table S4 (293065_1_supp_6572214_rpghjk.pdf).** *M. marinum* proteins shared between the *in vitro* biofilm surfaces identified using the biotinylation and cell-surface shaving (data from Table S3 and Savijoki et al., 2021) proteomics.

**Table S5 (293065_1_supp_6572215_rpghjk.pdf).** List of all mycobacterial proteins identified fron granulomas using LC-MS/MS. Both the original and normalized raw intensity values (normalized to combined raw intensity values of all identified proteins per replica) for each protein are shown. GroEL chaperones are shown with red letters.

**FIG S1 (293065_1_supp_6578168_rpf7yk.pdf).** Recombinant GroEL1 and GroEL2 production in *E. coli* as tag-free and AVI-tagged biotinylated form. (A) Final samples of GroEL1 protein in biotinylated and tag-free form were analysed via SDS-PAGE and Coomassie staining to verify purity. (B) The biotinylation of GroEL1 was verified by 1 h co-incubation at RT of the protein with tamavidin and SDS-PAGE (without boiling) to see the shift of the band upon the binding of tamavidin. The tag-free form was used as a negative control. (C) On the left, the tamavidin shift assay was performed with biotinylated and tag-free GroEL2 samples as in B. On the right, final samples of biotinylated and tag-free GroEL2 were analysed for purity. (D) A representative graph from the SEC carried out with a Superdex 200 column to verify monodispersity of the purified proteins. (E–H) The folding and thermal stability of GroEL1 and GroEL2 as biotinylated and tag-free forms were verified by nano-DSF in buffer corresponding to the screening conditions (25 mM Tris pH 8.0 and NaCl 150 mM). The inflection point temperatures are indicated with arrows.

**FIG S2 (293065_1_supp_6572312_rpghjk.pdf).** Verification of the protein identity of (A) GroEL1 and (B) GroEL2 by mass spectrometry (tryptic digest).

**FIG S3 (293065_1_supp_6578169_rpghjk.pdf).** Control experiments for sybody binding (A) As a negative control, 1 µg or 5 µg of His-Myc-tagged unspecific mannose binding protein (MBP) sybody was incubated with one-week-old Mmr biofilms. The sybodies bound to biofilm pellets were run onto SDS-page, and Western blots were stained with anti-Myc antibodies. No unspecific binding of this sybody was detected. On the first lane, 50 ng of MBP sybody was directly loaded onto the gel. (B-D) Two-week-old avirulent Mtb biofilms were stained with DAPI (shown in cyan) for DNA and with Myc-tagged sybodies against GroEL1 (C) or GroEL2 (D) + green-fluorescent anti-Myc antibodies (shown in

yellow) or DAPI and anti-Myc antibody as a negative control (B). The stained biofilms were imaged with confocal microscopy. The scale bars are 10 µm.

**Video S1 (293065_1_video_6578219_rp989w.mov).** 3D representation of GroEL1 sybody binding to granulomas. Pseudo-coloured 3D images of sybody binding to granulomas eight weeks post infection. DAPI (cyan) shows the host cell DNA, purple rods are mycobacteria and orange areas show the biofilm matrix stained with fluorescence-labelled sybodies.

**Video S2 (293065_1_video_6578244_rp888x.mov).** 3D representation of GroEL1 sybody binding to granulomas. Pseudo-coloured 3D images of sybody binding to granulomas eight weeks post infection. DAPI (cyan) shows the host cell DNA, purple rods are mycobacteria and orange areas show the biofilm matrix stained with fluorescence-labelled sybodies.

## Open Peer Review

**PEER REVIEW HISTORY (review-history.pdf).** An accounting of the reviewer comments and feedback.

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
