## [Reviewer comments · mSystems]

***In vitro* and *ex vivo* proteomics of *Mycobacterium marinum* biofilms and the development of biofilm-binding synthetic nanobodies**

Milka Hammarén, Hanna Luukinen, Alina Sillanpää, Kim Remans, Karine Lapouge, Tânia Custódio, Christian Löw, Henna Myllymäki, Toni Montonen, Markus Seeger, Joseph Robertson, Tuula Nyman, Kirsi Savijoki, and Matalena Parikka

Corresponding Author(s): Milka Hammarén, Tampereen yliopisto

Review Timeline:

Submission Date:	October 31, 2022
Editorial Decision:	December 21, 2022
Revision Received:	February 2, 2023
Accepted:	March 5, 2023

Editor: Ileana Cristea

Reviewer(s): Disclosure of reviewer identity is with reference to reviewer comments included in decision letter(s). The following individuals involved in review of your submission have agreed to reveal their identity: Paul Cos (Reviewer #1)

Transaction Report:

DOI: <https://doi.org/10.1128/msystems.01073-22>

December 21, 2022

Dr. Milka Marjut Hammarén
Tampereen yliopisto
Tampere
Finland

Re: mSystems01073-22 (*In vitro* and *ex vivo* proteomics of *Mycobacterium marinum* biofilms and the development of biofilm-binding synthetic nanobodies)

Dear Dr. Milka Marjut Hammarén:

Thank you for submitting your manuscript to mSystems. We have completed our review and I am pleased to inform you that, in principle, we expect to accept it for publication in mSystems. However, acceptance will not be final until you have adequately addressed all the reviewers' comments.

Preparing Revision Guidelines

Sincerely,

Ileana Cristea

Editor, mSystems

Journals Department
Reviewer comments:

Reviewer #1 (Comments for the Author):

This is an interesting manuscript studying the *Mycobacterium marinum* proteins of the biofilm matrices. The authors also created synthetic nanobodies (sybodies) against two mycobacterial chaperones as a potential tool for future therapies.

Major comments

1. In the introduction, the authors state that *Mycobacterium marinum* forms mature antibiotic-tolerant biofilms in vitro (ref 10) and is a well-established model pathogen for TB. These two statements are based on one publication, but they are essential to support the aim of the study. The authors should add some other references and discuss these two statements more in detail.
2. According to figure 1, a mycobacterial granuloma extracted from infected adult zebrafish contains a biofilm. However, many other molecules/cells that are not part of a biofilm are also present in a granuloma.
 - a. How can the authors be sure that a biofilm is formed? How do they check that in their experiments?
 - b. Lines 211-215: The authors state that "promoting the solubilisation of the mycobacterial proteins present either on the biofilm matrices or released into the granuloma environment". It is not clear to me how the authors can be sure that proteins from the biofilm were measured?
3. Line 162-176: In this part the abundance of surface proteins from biotinylated biofilms are compared to the ones of non-biotinylated biofilms. 432 proteins were found to be more abundant in biotinylated biofilm compared to non-biotinylated biofilms and these proteins were consequently chosen as potential surface-exposed targets on mycobacterial biofilms. Why can you conclude that these proteins are good targets based on the comparison between biotinylated and non-biotinylated biofilms? If both the biotinylated and non-biotinylated biofilms are grown in the same way, lysed before analysis and the only difference is in the biotinylation process (the process of covalently attaching biotin to a protein, nucleic acid or other molecule), should you not find the same amount of matrix-associated proteins in both conditions? Wouldn't this data only show that biotinylation is a better and more sensitive method to identify surface proteins? Why can you use non-biotinylated biofilms as a control here? Please explain this more in detail in the manuscript.

Minor comments

1. Line 182: "The LFQ data obtained by cell-surface tryptic shaving included the identifications from the planktonic cell surfaces after four days of growth 183 and from pellicle- and submerged-type biofilms after four weeks of growth at +28{degree sign}C." How can you compare the analysis made on planktonic cells after 4 days of growth with the one made on biofilms after 4 weeks of growth? Do you check if they are in a similar growth phase?
2. Line 288-290: Why do you use different staining methods at different timepoints?
3. Line 232 and 278: Maybe you could use the sybodies or green fluorophore method and confocal microscopy analysis to compare 2 days old - 1 week old biofilms with 2-3 weeks old biofilms to further confirm your findings on GroEL1 being more abundant in early stage biofilms.
4. Line 425: How do you know if the biofilms cultured on 7H10 agar are really biofilms and not simple colonies?

Typos and corrections

1. Several consecutive sentences throughout the manuscript include some repetitive words:
 - a. Line 71 and 73: "Antibiotic tolerance" and "Tolerance".
 - b. Line 113 and 114: ", in this study,".
 - c. Line 149 and 152: "In addition" and "An additional".
 - d. Line 253 and 254: "The purified proteins" and "The proteins".
 - e. Line 377: "Among".
 - f. Line 380, 382 and 384: "also".
 - g. Line 445, 446 and 450: "After".
 - h. Line 456 and 459: "The samples were".
 - i. Line 553 and 555: "The presence of".
 - j. Line 582 until 601: "The ... were" ("The bacteria were", "The cell pellets were", "The cells were", "The cleaned supernatans were", ...)
2. Line 98: ", in the future," -> put this differently in the sentence.
3. Line 115: "in Mmr granulomas" -> "of Mmr granulomas".
4. Line 170: "two to 14 TMDs" -> write both numbers in the same way.
5. Line 189: "with" -> "within".
6. Line 213: "disruption/lysis" -> same meaning, choose one.
7. Line 225: "The presence of GroEL1 and GroEL2" -> "The data obtained on GroEL1 and GroEL2".
8. Line 266: "list of seven and 16" -> write both numbers in the same way.
9. Line 288: "We saw" -> "Furthermore, we observed".
10. Line 313: "to bind inside granulomas" -> "to bind biofilms inside granulomas".
11. Line 370: "Although of interest for understanding" -> "Although it is interesting to understand".
12. Line 402: "This study provides" -> "As a conclusion, this study provides" to emphasize the fact that this part is a general conclusion of the manuscript will make it easier for the reader to read.
13. Line 484: "OD640" -> "OD600".
14. Line 496 and 497: "In the solution-digestion of zebrafish" is unclear.

15. Line 5556-558: This sentence is unclear.
16. Line 699: "generated during and/or analysed during" -> "generated and/or analysed during".
17. Line 942: "by the surface biotinylation" -> "by surface biotinylation"
18. Line 943: "followed by lysis, purification and proteomics." -> "followed by lysis, protein extraction and proteomic analysis." (It is easier for the reader when you stick to the words used in the figure.
19. Line 944: "and the total soluble" -> "of which the total soluble".
20. Line 947: "by western blotting" -> "and analysed by western blotting".
21. Line 967-972: "Mmr 967 biofilms were incubated with 0.1, 1 or 5 µg of GroEL1 sybodies or 1 or 5 µg of His-GroEL2 sybodies followed by washes of the biofilm pellets, boiling in sample buffer, SDS-PAGE and detection of the bound sybodies in the sample by western blot using anti-His (in GroEL1-SB experiments) or anti-myc (in GroEL2-SB experiments) antibodies." -> first ant-His GrEL2 sybodies are mentioned and then anti-myc, is this a mistake?

Reviewer #2 (Comments for the Author):

Hammaren and colleagues reported in a work performed to identify *M.marinum* ,as a surrogate for *M.tuberculosis*, proteins present in the biofilm matrix.

The authors used *M.marinum* to understand aspect of the *M.tb* biofilm. By performing proteomics proteomics of the matrix in vitro, the investigators found out that 160 proteins were expressed differently in biofilm matrix compared with proteins expressed by the planktonic phenotype. In addition, the investigators obtained granuloma tissue from *M.marinum* infected Zebrafish and examined the protein content of the biofilm ex-vivo.

Among the 91 proteins identified, GroEL1 and GroEL2 were the ones with greater expression in vitro and in vivo. By using fluorescent microscopy and sybodies (synthetic nanobodies) against the proteins, the investigators confirmed their presence on the surface matrix.

Comments and Suggestions

1. The techniques employed by the investigators were adequate for the purpose. The fact that many of the enriched proteins were enzymes, one wonder if no contamination with bacterial proteins did not occur. The authors should at least, discuss the fact.
2. The data is interesting, but how do the authors explain the absence of structural proteins in the biofilm? Are the enzymes and stress related proteins the difference between biofilm and planktonic phenotypes? Have the authors considering treating the biofilm with trypsin?
3. Figure 3. Some extra bands in the Western blots, appear to bind some of the antibodies. The authors should discuss the finding.
4. Figure 4. In several of the panels, the antibody binding seems to be localized to regions of the biofilm. Do the authors have an explanation for it?

RESPONSE to REVIEWERS' comments

Dear Reviewers, Dear Editor,

As a corresponding author, I want to thank the Reviewers for their insightful comments and feedback aiming at further improving the quality of our manuscript submitted to mSystems.

Please find below our point-by-point responses to the reviewers' comments and detailed description of the accompanying changes in the revised manuscript.

The page and line numberings in the comments refer to the original manuscript, and the numberings in our answers refer to the revised marked-up manuscript file.

The comments and our responses:

Reviewer #1 (Comments for the Author):

This is an interesting manuscript studying the *Mycobacterium marinum* proteins of the biofilm matrices. The authors also created synthetic nanobodies (sybodies) against two mycobacterial chaperones as a potential tool for future therapies.

Major comments

1. In the introduction, the authors state that *Mycobacterium marinum* forms mature antibiotic-tolerant biofilms in vitro (ref 10) and is a well-established model pathogen for TB. These two statements are based on one publication, but they are essential to support the aim of the study. The authors should add some other references and discuss these two statements more in detail.

Author response: M. marinum, similarly to other relevant non-tuberculous mycobacteria (reviewed in Chakraborty and Kumar 2019) is known to form biofilms in vitro (Hall-Stoodley 2006, Ren et al. 2007, Savijoki et al. 2021). In addition, the mycobacterial biofilms have been shown to be tolerant to antibiotics in vitro (Chakraborty et al. 2021, Savijoki et al. 2021) and in vivo (Chakraborty et al. 2021).

2. According to figure 1, a mycobacterial granuloma extracted from infected adult zebrafish contains a biofilm. However, many other molecules/cells that are not part of a biofilm are also present in a granuloma.

a. How can the authors be sure that a biofilm is formed? How do they check that in their experiments?

Author response: Biofilms have been shown to exist inside granulomas in various animal models (mouse, macaque, rabbit) (Chakraborty et al. 2021). In vivo biofilm detection is based on cellulose staining. Cellulose which is a hallmark of mycobacterial biofilm and cannot be produced into the lesion by the host. In our group, we have additionally used HPLC-MS mass-spectrometry on enzyme treated mycobacterial samples to measure the cellulose content and hence verify the biofilm formation by M. marinum.

b. Lines 211-215: The authors state that "promoting the solubilisation of the mycobacterial proteins

present either on the biofilm matrices or released into the granuloma environment". It is not clear to me how the authors can be sure that proteins from the biofilm were measured?

Author response: *This statement is based on our previously published study, in which we compared the cell-surface associated proteins of M. marinum grown in planktonic/single cell state with those associated on the mature biofilms at different timepoints of biofilm development. Since the mycobacterial granulomas were dissected at the late timepoint of infection (8-wpi) and previous report by Chakraborty demonstrated that mycobacteria form biofilms in vivo, it is highly likely that also M. marinum had switched into a biofilm mode of growth in granulomas after 8 weeks to prolong its viability. The main idea of the present study was to confirm that the selected target proteins captured by in vitro biotinylation could also be identified with reasonably high raw intensity values from granulomas, which then suggests that the proteins in question are present in detectable amounts in vivo and could be used to develop nanobodies. We agree with the referee in that some proteins may have originated also from M. marinum cells present in single cell state. Since no method to isolate biofilm and planktonic cells separately from granulomas is yet available, we have added necessary discussion related to the presence of non-biofilm proteins within the identified protein list into the relevant part of the manuscript (page 10, lines 234-237).*

3. Line 162-176: In this part the abundancy of surface proteins from biotinylated biofilms are compared to the ones of non-biotinylated biofilms. 432 proteins were found to be more abundant in biotinylated biofilm compared to non-biotinylated biofilms and these proteins were consequently chosen as potential surface-exposed targets on mycobacterial biofilms. Why can you conclude that these proteins are good targets based on the comparison between biotinylated and non-biotinylated biofilms? If both the biotinylated and non-biotinylated biofilms are grown in the same way, lysed before analysis and the only difference is in the biotinylation process (the process of covalently attaching biotin to a protein, nucleic acid or other molecule), should you not find the same amount of matrix-associating proteins in both conditions? Wouldn't this data only show that biotinylation is a better and more sensitive method to identify surface proteins? Why can you use non-biotinylated biofilms as a control here? Please explain this more in detail in the manuscript.

Author response: *When membrane impermeable Sulfo-NHS-LC-biotin labelling is applied to intact biofilms, the biotin label becomes attached to the externally available free amino groups on the exposed proteins. The deeper the protein is localized, the less likely it is to become labelled. Hence, when pulled down with streptavidin beads, the collected protein pool is enriched with surface-exposed proteins. As negative controls, the non-biotinylated control samples were not treated with Sulfo-NHS-LC-biotin but were lysed similarly to the biotinylated sample and the proteins were pulled down with streptavidin beads in separate tubes. The streptavidin pull-down from the non-biotinylated samples shows the level of non-specific binding to the streptavidin beads/tube and the endogenously biotinylated proteins. Hence, if a certain protein was also pulled down in the non-biotinylated sample to the same extent as in the surface-biotinylated sample type, it was removed from the hit list, as its presence in the sample was likely to be due to non-specific binding. Proteins that were more abundant in the pulldown, were identified as likely surface-exposed proteins. The experimental setup is now explained more clearly, and the above-mentioned information is added into the results section of the manuscript (page 6-7, lines 143-156).*

Minor comments

1. Line 182: "The LFQ data obtained by cell-surface tryptic shaving included the identifications from the planktonic cell surfaces after four days of growth and from pellicle- and submerged-type biofilms after four weeks of growth at +28 °C." How can you compare the analysis made on planktonic cells after 4 days of growth with the one made on biofilms after 4 weeks of growth? Do you check if they are in a similar growth phase?

Author response: *The tryptic shaving data has already been previously published in mSystems (Savijoki et al. 2021). Evidently, a mature biofilm and a planktonic culture at 4 days are in an entirely different growth phase. At 4-day timepoint, the bacteria are actively growing and dividing, whereas after 4 weeks, bacteria have switched into a biofilm mode of growth to prolong the viability of the residing cells and help the biofilm cells to resist e.g., antibiotics and host immune response. The main interest of such a comparison was to study the differences between these two growth phases were at the proteome level.*

2. Line 288-290: Why do you use different staining methods at different timepoints?

Author response: *Different staining methods were used due to practical reasons. When processing the first samples (at two-week timepoint), we did not have the directly labeled nanobodies available yet, and therefore used myc-antibody instead. The directly labeled nanobodies became available later, which were then used in the following experiments. Both staining strategies worked comparably.*

3. Line 232 and 278: Maybe you could use the sybodies or green fluorophore method and confocal microscopy analysis to compare 2 days old - 1 week old biofilms with 2-3 weeks old biofilms to further confirm your findings on GroEL1 being more abundant in early stage biofilms.

Author response: *We do not aim to make specific claims on the abundance of GroEL1 in early vs. late-stage biofilms based on the proteomic results shown in this study. The important information extracted is that GroEL1 is present and abundant on the surface of mycobacterial biofilms in vitro, and that this chaperone can also be detected with reasonably high raw intensity values after LC-MS/MS analysis from granulomas withdrawn after 8 wpi. The kinetics of biofilm formation and the relative abundance of GroEL1 has been previously assessed by Savijoki et al. 2021.*

4. Line 425: How do you know if the biofilms cultured on 7H10 agar are really biofilms and not simple colonies?

Author response: *We have not assessed if the selected M. marinum strain grows as a biofilm on 7H10 agar, since biofilms for the present study were formed in liquid cultures.*

Typos and corrections

1. Several consecutive sentences throughout the manuscript include some repetitive words:

- a. Line 71 and 73: "Antibiotic tolerance" and "Tolerance".
- b. Line 113 and 114: ", in this study,".
- c. Line 149 and 152: "In addition" and "An additional".
- d. Line 253 and 254: "The purified proteins" and "The proteins".
- e. Line 377: "Among".
- f. Line 380, 382 and 384: "also".
- g. Line 445, 446 and 450: "After".
- h. Line 456 and 459: "The samples were".
- i. Line 553 and 555: "The presence of".

Author response: *the words indicated above (1a – 1i) were corrected as requested by the*

referee.

j. Line 582 until 601: "The ... were" ("The bacteria were", "The cell pellets were", "The cells were", "The cleaned supernatans were", ...)

Author response: we decided to leave these words/sentences unchanged as advised by the native language corrected.

2. Line 98: ", in the future," -> put this differently in the sentence.

3. Line 115: "in Mmr granulomas" -> "of Mmr granulomas".

Author response: the sentences indicated above (2-3) were corrected as requested by the referee.

4. Line 170: "two to 14 TMDs" -> write both numbers in the same way.

Author response: We were advised to leave this sentence unchanged according to the native language; numbers below ten should be spelled out.

5. Line 189: "with" -> "within".

6. Line 213: "disruption/lysis" -> same meaning, choose one.

7. Line 225: "The presence of GroEL1 and GroEL2" -> "The data obtained on GroEL1 and GroEL2".

Author response: the words/sentences indicated above (5-7) were corrected as requested by the referee.

8. Line 266: "list of seven and 16" -> write both numbers in the same way.

Author response: We were advised to leave this sentence unchanged according to the native language; numbers below ten should be spelled out.

9. Line 288: "We saw" -> "Furthermore, we observed".

10. Line 313: "to bind inside granulomas" -> "to bind biofilms inside granulomas".

11. Line 370: "Although of interest for understanding" -> "Although it is interesting to understand".

12. Line 402: "This study provides" -> "As a conclusion, this study provides" to emphasize the fact that this part is a general conclusion of the manuscript will make it easier for the reader to read.

13. Line 484: "OD640" -> "OD600".

14. Line 496 and 497: "In the solution-digestion of zebrafish" is unclear.

Author response: the words/sentences indicated above (9-14) were corrected as requested by the referee

15. Line 5556-558: This sentence is unclear.

Author response: This is a bioinformatics tool used for predicting secretion motifs on proteins.

16. Line 699: "generated during and/or analysed during" -> "generated and/or analysed during".

17. Line 942: "by the surface biotinylation" -> "by surface biotinylation"

18. Line 943: "followed by lysis, purification and proteomics." -> "followed by lysis, protein extraction and proteomic analysis." (It is easier for the reader when you stick to the words used in the figure.

19. Line 944: "and the total soluble" -> "of which the total soluble".

20. Line 947: "by western blotting" -> "and analysed by western blotting".

Author response: the words/sentences indicated above (16-20) were corrected as requested

by the referee.

21. Line 967-972: "Mmr 967 biofilms were incubated with 0.1, 1 or 5 µg of GroEL1 sybodies or 1 or 5 µg of His-GroEL2 sybodies followed by washes of the biofilm pellets, boiling in sample buffer, SDS-PAGE and detection of the bound sybodies in the sample by western blot using anti-His (in GroEL1-SB experiments) or anti-myc (in GroEL2-SB experiments) antibodies." -> first ant-His GrEL2 sybodies are mentioned and then anti-myc, is this a mistake?

Author response: No, we could use both antibodies for detection, as the nanobodies contained two tags.

Reviewer #2 (Comments for the Author):

Hammaren and colleagues reported in a work performed to identify *M. marinum* ,as a surrogate for *M. tuberculosis*, proteins present in the biofilm matrix.

The authors used *M. marinum* to understand aspect of the *M. tb* biofilm. By performing proteomics proteomics of the matrix in vitro, the investigators found out that 160 proteins were expressed differently in biofilm matrix compared with proteins expressed by the planktonic phenotype. In addition, the investigators obtained granuloma tissue from *M. marinum* infected Zebrafish and examined the protein content of the biofilm ex-vivo.

Among the 91 proteins identified, GroEL1 and GroEL2 were the ones with greater expression in vitro and in vivo. By using fluorescent microscopy and sybodies (synthetic nanobodies) against the proteins, the investigators confirmed their presence on the surface matrix.

Comments and Suggestions

1. The techniques employed by the investigators were adequate for the purpose. The fact that many of the enriched proteins were enzymes, one wonder if no contamination with bacterial proteins did not occur. The authors should at least, discuss the fact.

Author response: We assume that the referee refers to intracellular enzymes being as the contaminating enzymes within the identified/enriched biofilm matrix proteins. These enzymes are commonly referred to as proteins with moonlighting function, which are expected to have entered the cell surface either by controlled cell lysis or yet by unknown mechanisms, both termed as "non-classical secretory pathway". Such proteins originating from different organisms, including bacteria, are currently available at the MoonProt database (Chen et al., 2020). Number of studies have demonstrated the presence of cytoplasmic proteins at the cell surface on different bacteria and on bacterial cells during planktonic and biofilm growth, and that their export is expected involve controlled cell lysis (involving autolysins, phenol-soluble modulins, membrane vesicles and/or secreted phages) (Antikainen et al., 2007; Kainulainen et al., 2012; Kainulainen and Korhonen, 2014; Espino et al., 2015; Graf et al., 2019; Harvey et al., 2019; Dengler et al., 2015; Götz et al., 2015; Ebner et al., 2015, 2016, 2017, 2019; Jeffery, 2019; Savijoki et al., 2011, 2019; Hiltunen et al., 2019; Hemmadi and Biswas, 2020; Reigada et al., 2021; San Martin-Galindo et al., 2021). Such proteins have also been identified from *Mycobacterium bovis* and *M. marinum* (Pagani et al., 2019; Savijoki et al., 2021). In addition, many cytoplasmic moonlighters have been demonstrated to show pH-dependent association with the cell-surface/biofilm matrix components; as soon as the pH of the surrounding environment drops, a large group of the

released cytoplasmic proteins become attached to the cell wall/biofilm matrix components (Foulston et al., 2014). This indicates that bacteria can control the attachment of the cytoplasmic proteins during physiological growth. Discussion related to the presence of cytoplasmic proteins and possible mechanisms controlling their release into the biofilm matrix is now rephrased with more supporting articles into the discussion (pages 16-18, lines 384-430).

2. The data is interesting, but how do the authors explain the absence of structural proteins in the biofilm?

Author response: *We assume that the referee refers to the cell-wall attached/anchored proteins carrying both the signal peptide targeting the protein to the classical secretory pathway and the signal motif(s) retaining the protein to the cell-wall. Here, the cytoplasmic proteins formed the major group of cell-surface attached proteins on the biofilm matrices, which is in line with several similar studies on other gram-positive bacteria, as stated above. It is also known that highly abundant proteins tend to mask the low- and ultralow-abundance proteins, making the low-abundance proteins difficult to identify by shot-gun proteomics. In this study, the protein identification method favored identification of the most abundant proteins that were easily assessed by biotinylation and streptavidin capturing, which explains why majority of the identified proteins were the cytoplasmic moonlighters, since these were detected with the highest raw intensity values. Structural proteins frequently contain highly hydrophobic and complex regions due to e.g., varying number of transmembrane spanning domains, which typically are difficult to identify due to the inherent lack of trypsin cleavage sites (K or R) in the hydrophobic regions of these proteins and to the tendency of the hydrophobic peptides to aggregate and to precipitate in aqueous solutions in the absence of solubilizing detergents. Since the poor signal obtained with hydrophobic peptides using conventional soft ionization techniques such as ESI in mass spectrometry analysis still contributes to difficulties in identifying such peptides, we assume that although our study allowed identification of some cell-wall/-membrane anchored proteins, detection of many other proteins of this group must have remained below detection limit due to the reasons stated above.*

*On the other hand, the cytoplasmic moonlighters, such as the ribosomal proteins (r-proteins), were identified as the most abundant protein group. As already mentioned, bacteria are proposed to recycle conserved cytoplasmic proteins in a pH-dependent manner into the matrix (Foulston et al., 2014), among which the r-proteins, by mediating strong positive charge, has been proposed to play a key role in stabilizing the biofilm structure and matrix in a gram-positive *Staphylococcus aureus* (Graf et al., 2019). Thus, we suggest that many of the identified moonlighters, also including r-proteins, could also strengthen the biofilm matrix/structure of *M. marinum*, as seen in *S. aureus*. Growing evidence from different bacterial biofilms has also linked the presence of fibrous proteins, including amyloids or amyloid-like fibers, to a functional role within the biofilm matrix; functional amyloids have been reported to serve as building blocks and provide mechanical robustness to the biofilm (Serra et al., 2013; Erskine et al., 2018). Due to their highly ordered beta-sheet rich filamentous morphology, an ability to interact erroneously and generate insoluble/protease-resistant aggregates/fibrils (Jain and Chapman, 2019), the protein amyloids most likely have remained unidentified in our study. This is because identification of such proteins requires dispersing/solubilizing the aggregates/fibrils with solvents, such as trifluoroacetic acid (TFA) or formic acid (FA) prior to tryptic digestion and LC-MS/MS identification (Kaneko et al., 2014). Since the proteomic identification method requires*

capturing the proteins in their native form, the use of these solvents during biotinylation would have affected the protein structure. Moreover, TFA or FA are typically used to inactive tryptic digestions, therefore their application before the tryptic digestion is not possible. This information is now added into the discussion section (pages 16-18, lines 384-430).

Are the enzymes and stress related proteins the difference between biofilm and planktonic phenotypes? Have the authors considering treating the biofilm with trypsin?

Author response: Yes, we have already performed such study that was published in mSystems by Savijoki et al. (2021). In that study, we compared the planktonic cell surfaces of M. marinum with M. marinum cells during biofilm mode of growth from 2 days to 3 months using cell surface shaving (combination of trypsin and endoproteinase rLysC) and LFQ identification. Several enzymes were found to show growth mode-dependent abundance change. For example, we observed that cellulase, the key factor controlling the cellulose-mediated biofilm formation, was highly abundant on planktonic cell surfaces and in the early stages of the biofilm formation. The abundance of this enzyme was decreased significantly in 2-week-old biofilms. Cellulose is one of the main components of the biofilm matrix in M. tuberculosis and our findings with M. marinum show that also in this species decreased cellulase activity is required to establish biofilm growth. In addition, we also observed that several other cytoplasmic moonlighters, e.g., GroEL1, show growth mode-dependent changes. All this data has been presented in Savijoki et al. (2021), and, in terms of GroEL1, is already presented in the manuscript (page 11, lines 249-256).

3. Figure 3. Some extra bands in the Western blots, appear to bind some of the antibodies. The authors should discuss the finding.

Author response: We apologize for not having this issue clarified in the manuscript. Indeed, there were extra bands detected on the Western blots, which most likely refer to the presence of the nanobody migrating as a partially unfolded state due to the experimental conditions used and the presence of intact disulfide bonds mediating nanobody-nanobody interactions. The samples and gel electrophoresis were conducted under non-reducing conditions and the antibodies in question harbor cysteines with free sulfhydryl sidechains capable of forming disulfide bonds. The formation of disulfide bonds between the individual nanobodies could explain why we detected several extra bands migrating with higher molecular weight in comparison to the monomeric nanobody. Similar findings have been detected with other proteins having disulfide bonds, with milk whey protein as an example (Pizzano et al., 2012). We have now explained the western blot results more accurately in the results section (page 13, lines 298-305).

4. Figure 4. In several of the panels, the antibody binding seems to be localized to regions of the biofilm. Do the authors have an explanation for it?

Author response: The binding pattern is expected to reflect the actual protein localization pattern in the samples. The protein may not be uniformly distributed on the biofilms.

References:

Antikainen J, Kuparinen V, Lähteenmäki K, Korhonen TK. pH-dependent association of enolase and glyceraldehyde-3-phosphate dehydrogenase of *Lactobacillus crispatus* with the cell wall and lipoteichoic acids. *J Bacteriol.* 2007. 189(12):4539-43. doi:10.1128/JB.00378-07.

Chakraborty P, Bajeli S, Kaushal D, Radotra BD, Kumar A. Biofilm formation in the lung contributes to virulence and drug tolerance of *Mycobacterium tuberculosis*. *Nat Commun.* 2021 11;12(1):1606. doi:10.1038/s41467-021-21748-6.

Chen C, Liu H, Zabad S, Rivera N, Rowin E, Hassan M, Gomez De Jesus SM, Llinás Santos PS, Kravchenko K, Mikhova M, Ketterer S, Shen A, Shen S, Navas E, Horan B, Raudsepp J, Jeffery C. MoonProt 3.0: an update of the moonlighting proteins database. *Nucleic Acids Res.* 2021. 8;49(D1):D368-D372. doi:10.1093/nar/gkaa1101.

Dunker AK, Kenyon AJ. Mobility of sodium dodecyl sulphate - protein complexes. *Biochem J.* 1976. 1;153(2):191-7. doi: 10.1042/bj1530191.

Ebner P, Prax M, Nega M, Koch I, Dube L, Yu W, Rinker J, Popella P, Flötenmeyer M, Götz F. Excretion of cytoplasmic proteins (ECP) in *Staphylococcus aureus*. *Mol Microbiol.* 2015. 97(4):775-89. doi:10.1111/mmi.13065.

Ebner P, Rinker J, Götz F. Excretion of cytoplasmic proteins in *Staphylococcus* is most likely not due to cell lysis. *Curr Genet.* 2016. 62(1):19-23. doi:10.1007/s00294-015-0504-z.

Ebner P, Luqman A, Reichert S, Hauf K, Popella P, Forchhammer K, Otto M, Götz F. Non-classical Protein excretion is boosted by PSM α -Induced cell leakage. *Cell Rep.* 2017. 8;20(6):1278-1286. doi:10.1016/j.celrep.2017.07.045.

Ebner P, Götz F. Bacterial excretion of cytoplasmic proteins (ECP): Occurrence, mechanism, and function. *Trends Microbiol.* 2019. 27(2):176-187. doi:10.1016/j.tim.2018.10.006.

Erskine E, MacPhee CE, Stanley-Wall NR. Functional Amyloid and Other Protein Fibers in the Biofilm Matrix. *J Mol Biol.* 2018 Oct 12;430(20):3642-3656. doi: 10.1016/j.jmb.2018.07.026.

Espino E, Koskeniemi K, Mato-Rodriguez L, Nyman TA, Reunanen J, Koponen J, Öhman T, Siljamäki P, Alatossava T, Varmanen P, Savijoki K. Uncovering surface-exposed antigens of *Lactobacillus rhamnosus* by cell shaving proteomics and two-dimensional immunoblotting. *J Proteome Res.* 2015 Feb 6;14(2):1010-24.

Foulston L, Elsholz AKW, DeFrancesco AS, Losick R. The extracellular matrix of *Staphylococcus aureus* biofilms comprises cytoplasmic proteins that associate with the cell surface in response to decreasing pH. *mBio.* 2014. 2;5(5):e01667-14. doi:10.1128/mBio.01667-14.

Hall-Stoodley L, Brun OS, Polshyna G, Barker LP. 2006. *Mycobacterium marinum* biofilm formation reveals cording morphology. *FEMS Microbiol Lett* 257:43–9.

Hiltunen AK, Savijoki K, Nyman TA, Miettinen I, Ihalainen P, Peltonen J, Fallarero A. Structural and Functional Dynamics of *Staphylococcus aureus* Biofilms and Biofilm Matrix Proteins on Different Clinical Materials. *Microorganisms.* 2019 7(12):584. doi:10.3390/microorganisms7120584.

Graf AC, Leonard A, Schäuble M, Rieckmann LM, Hoyer J, Maass S, Lalk M, Becher D, Pané-Farré J, Riedel K. Virulence Factors Produced by *Staphylococcus aureus* biofilms have a moonlighting function contributing to biofilm integrity. *Mol Cell Proteomics.* 2019. 18(6):1036-1053. doi: 10.1074/mcp.RA118.001120.

Götz F, Yu W, Dube L, Prax M, Ebner P. Excretion of cytosolic proteins (ECP) in bacteria. *Int J Med Microbiol.* 2015. 305(2):230-7. doi:10.1016/j.ijmm.2014.12.021.

Hemmadi V and Biswas M. An overview of moonlighting proteins in *Staphylococcus aureus* infection. *Arch Microbiol.* 2020. 13 : 1–18. doi:10.1007/s00203-020-02071-y.

Jain N and Chapman MR. Bacterial functional amyloids: Order from disorder. *Biochim Biophys Acta Proteins Proteom,* 2019. 1867, 954-960. doi: 10.1016/j.bbapap.2019.05.010

Jeffery CJ. Intracellular/surface moonlighting proteins that aid in the attachment of gut microbiota to the host. *AIMS Microbiology.* 2019, 5, 1: 77-86. doi: 10.3934/microbiol.2019.1.77.

Kainulainen V, Korhonen TK. Dancing to another tune-adhesive moonlighting proteins in bacteria. *Biology (Basel)*. 2014. 10;3(1):178-204. doi:10.3390/biology3010178.

Kainulainen V, Loimaranta V, Pekkala A, Edelman S, Antikainen J, Kylväjä R, Laaksonen M, Laakkonen L, Finne J, Korhonen TK. Glutamine synthetase and glucose-6-phosphate isomerase are adhesive moonlighting proteins of *Lactobacillus crispatus* released by epithelial cathelicidin LL-37. *J Bacteriol*. 2012. 194(10):2509-19. doi:10.1128/JB.06704-11.

Kaneko N, Yamamoto R, Sato TA, Tanaka K. Identification and quantification of amyloid beta-related peptides in human plasma using matrix-assisted laser desorption/ionization time-of-flight mass spectrometry. *Proc Jpn Acad Ser B Phys Biol Sci*. 2014;90(3):104-117. doi:10.2183/pjab.90.104.

Pagani TD, Guimarães ACR, Waghbi MC, Corrêa PR, Kalume DE, Berrêdo-Pinho M, Degraive WM, Mendonça-Lima L. Exploring the potential role of moonlighting function of the surface-associated proteins from *Mycobacterium bovis* BCG Moreau and Pasteur by comparative proteomic. *Front Immunol*. 2019. 26;10:716. doi:10.3389/fimmu.2019.00716.

Pitt-Rivers R, Impiombato FS. The binding of sodium dodecyl sulphate to various proteins. *Biochem J*. 1968, 109(5):825-30. doi: 10.1042/bj1090825.

Pizzano R, Manzo C, Nicolai MA, Addeo F. Occurrence of major whey proteins in the pH 4.6 insoluble protein fraction from UHT-treated milk. *J Agric Food Chem*. 2012 Aug 15;60(32):8044-50. doi: 10.1021/jf3024563.

Reigada I, San-Martin-Galindo P, Gilbert-Girard S, Chiaro J, Cerullo V, Savijoki K, Nyman TA, Fallarero A, Miettinen I. Surfaceome and Exoproteome Dynamics in Dual-Species *Pseudomonas aeruginosa* and *Staphylococcus aureus* Biofilms. *Front Microbiol*. 2021. doi: 10.3389/fmicb.2021.672975.

Ren H, Dover LG, Islam ST, Alexander DC, Chen JM, Besra GS, Liu J. 2007. Identification of the lipooligosaccharide biosynthetic gene cluster from *Mycobacterium marinum*. *Mol Microbiol* 63:1345–1359.

Sani M, Houben EN, Geurtsen J, Pierson J, de Punder K, van Zon M, Wever B, Piersma SR, Jiménez CR, Daffé M, Appelmelk BJ, Bitter W, van der Wel N, Peters PJ. Direct visualization by cryo-EM of the mycobacterial capsular layer: a labile structure containing ESX-1-secreted proteins. *PLoS Pathog*. 2010. 5;6(3):e1000794. doi:10.1371/journal.ppat.1000794.

San-Martin-Galindo P, Rosqvist E, Tolvanen S, Miettinen I, Savijoki K, Nyman TA, Fallarero A, Peltonen J. Modulation of virulence factors of *Staphylococcus aureus* by nanostructured surfaces, *Materials & Design*, 208, 2021, 109879. doi:10.1016/j.matdes.2021.109879.

Savijoki K, Iivanainen A, Siljamäki P, Laine PK, Paulin L, Karonen T, Pyörälä S, Kankainen M, Nyman TA, Salomäki T, Koskinen P, Holm L, Simojoki H, Taponen S, Sukura A, Kalkkinen N, Auvinen P, Varmanen P. Genomics and Proteomics Provide New Insight into the Commensal and Pathogenic Lifestyles of Bovine- and Human-Associated *Staphylococcus epidermidis* Strains. *J Proteome Res*. 2014 Aug 1;13(8):3748-3762. doi: 10.1021/pr500322d

Savijoki K, Miettinen I, Nyman TA, Kortesoja M, Hanski L, Varmanen P, Fallarero A. Growth Mode and Physiological state of cells prior to biofilm formation affect immune evasion and persistence of *Staphylococcus aureus*. *Microorganisms*. 2019. 12;8(1):106. doi: 10.3390/microorganisms8010106.

Savijoki K, Myllymäki H, Luukinen H, Paulamäki L, Vanha-Aho LM, Svorjova A, Miettinen I, Fallarero A, Ihalainen TO, Yli-Kauhaluoma J, Nyman TA, Parikka M. Surface-Shaving Proteomics of *Mycobacterium marinum* Identifies Biofilm Subtype-Specific Changes Affecting Virulence, Tolerance, and Persistence. *mSystems*. 2021 Jun 29;6(3):e0050021. doi: 10.1128/mSystems.00500-21.

Serra DO, Richter AM, Klauck G, Mika F, Hengge R. Microanatomy at cellular resolution and spatial order of physiological differentiation in a bacterial biofilm. *mBio*. 2013 4(2):e00103-13. doi: 10.1128/mBio.00103-13.

Van Wyk N, Navarro D, Blaise M, Berrin J-G, Henrissat B, Drancourt M, Kremer L. Characterization of a mycobacterial cellulase and its impact on biofilm- and drug-induced cellulose production. *Glycobiology*. 2017. 1;27(5):392-399. doi:10.1093/glycob/cwx014.

March 5, 2023

Dr. Milka Marjut Hammarén
Tampereen yliopisto
Tampere
Finland

Re: mSystems01073-22R1 (*In vitro* and *ex vivo* proteomics of *Mycobacterium marinum* biofilms and the development of biofilm-binding synthetic nanobodies)

Dear Dr. Milka Marjut Hammarén:

Congratulations!

Your manuscript has been accepted, and I am forwarding it to the ASM Journals Department for publication. For your reference, ASM Journals' address is given below. Before it can be scheduled for publication, your manuscript will be checked by the mSystems production staff to make sure that all elements meet the technical requirements for publication. They will contact you if anything needs to be revised before copyediting and production can begin. Otherwise, you will be notified when your proofs are ready to be viewed.

If you would like to submit a potential Featured Image, please email a file and a short legend to msystems@asmusa.org. Please note that we can only consider images that (i) the authors created or own and (ii) have not been previously published. By submitting, you agree that the image can be used under the same terms as the published article. File requirements: square dimensions (4" x 4"), 300 dpi resolution, RGB colorspace, TIF file format.

We recognize that the video files can become quite large, and so to avoid quality loss ASM suggests sending the video file via <https://www.wetransfer.com/>. When you have a final version of the video and the still ready to share, please send it to mSystems staff at msystems@asmusa.org.

Sincerely,

Ileana Cristea
Editor, mSystems

Journals Department
E-mail: mSystems@asmusa.org